# Training Data Attribution via Approximate Unrolling

**Juhan Bae**[1,2], **Wu Lin**[2], **Jonathan Lorraine**[1,2,3], **Roger Grosse**[1,2,4]

[1]University of Toronto, [2]Vector Institute, [3]NVIDIA, [3]Anthropic

{jbae, lorraine, rgrosse}@cs.toronto.edu
wu.lin@vectorinstitute.ai

## Abstract

Many training data attribution (TDA) methods aim to estimate how a model's behavior would change if one or more data points were removed from the training set. Methods based on implicit differentiation, such as influence functions, can be made computationally efficient, but fail to account for underspecification, the implicit bias of the optimization algorithm, or multi-stage training pipelines. By contrast, methods based on unrolling address these issues but face scalability challenges. In this work, we connect the implicit-differentiation-based and unrolling-based approaches and combine their benefits by introducing SOURCE, an approximate unrolling-based TDA method that is computed using an influence-function-like formula. While being computationally efficient compared to unrolling-based approaches, SOURCE is suitable in cases where implicit-differentiation-based approaches struggle, such as in non-converged models and multi-stage training pipelines. Empirically, SOURCE outperforms existing TDA techniques in counterfactual prediction, especially in settings where implicit-differentiation-based approaches fall short.

## 1 Introduction

Training data attribution (TDA) techniques are motivated by understanding the relationship between training data and the properties of trained models [92, 17, 29, 35, 70, 24, 51]. They have diverse applications in machine learning, such as detecting mislabeled data points [72, 50, 41], crafting data poisoning attacks [16, 38, 69], and curating datasets [60, 90, 13]. Many TDA methods aim to perform a *counterfactual prediction*, which estimates how a trained model's behavior would change if certain data points were removed from (or added to) the training dataset. Unlike sampling-based approaches, which require repeated model retraining with different subsets of the dataset, gradient-based TDA techniques estimate an infinitesimal version of the counterfactual without model retraining. Two main strategies for gradient-based counterfactual TDA are *implicit differentiation* and *unrolling*.

Implicit-differentiation-based TDA, most notably influence functions [28, 49], uses the Implicit Function Theorem [52] to estimate the sensitivity of the optimal solution to downweighting a training data point. These methods are well-motivated for models with strongly convex objectives and provide convenient estimation algorithms that depend solely on the optimal model parameters rather than intermediate checkpoints throughout training. However, the classical formulation relies on assumptions such as the uniqueness of and convergence to the optimal solution, which limits their applicability to modern neural networks [6, 3, 77].

By contrast, unrolling-based methods, such as SGD-INFLUENCE [31], approximate the impact of downweighting a data point's gradient update on the final model parameters by backpropagating through the preceding optimization steps. Unrolling is conceptually appealing in modern neural networks because it does not rely on the uniqueness of or convergence to the optimal solution. Furthermore, it can incorporate details of the training process, such as the choice of optimizer, learning rate schedules, or a data point's position during training. For example, unrolling-based

approaches can support TDA for multi-stage training procedures, such as in continual learning or foundation models, where the model undergoes multiple training phases with different objectives or datasets. However, they require storing all intermediate variables generated during the training process (*e.g.*, parameter vectors for each optimization step) for backpropagation, which can be prohibitively expensive for large-scale models. Past works have considered applying unrolling to only the last epoch for large-scale models [31, 10], restricting their applicability in analyzing the effect of removing a data point at the beginning of training or in analyzing multi-stage training procedures.

In this work, we connect implicit-differentiation-based and unrolling-based approaches and introduce a novel algorithm that enjoys the advantages of both methods. We start from the unrolled differentiation perspective and, after introducing suitable approximations, arrive at an influence-function-like estimation algorithm. The key idea is to divide the training trajectory into one or more segments and approximate the distributions of gradients and Hessians as stationary within each segment. These segments may represent explicit training stages, such as in continual learning or foundation models, or changes in the Hessian and gradients throughout training. We use these estimated statistical summaries for each segment to approximate unrolling. Hence, we call our method SOURCE (**S**egmented stati**O**nary **U**n**R**olling for **C**ounterfactual **E**stimation). While our method approximately coincides with influence functions in the simple setting of a deterministic objective optimized to convergence, it applies to more general settings where unrolling is typically required.

SOURCE inherits several key advantages from unrolling. Firstly, it allows the attribution of data points at different stages of training, providing a more comprehensive framework for TDA. Secondly, SOURCE can incorporate algorithmic choices into the analysis, accounting for learning rate schedules and the implicit bias of optimizers such as SGD or Adam [48]. Lastly, it maintains a close connection with the counterfactuals, even in cases where the assumptions made in implicit-differentiation-based methods, such as the optimality of the final parameters, are not met. However, unlike unrolling, SOURCE does not require storing all intermediate optimization variables generated during training; instead, it leverages only a handful of model checkpoints.

We evaluate SOURCE for counterfactual prediction across various tasks, including regression, image classification, and text classification. SOURCE outperforms existing TDA techniques in approximating the effect of retraining the network without groups of data points and identifying training data points that would flip predictions on some test examples when trained without them. SOURCE demonstrates distinct advantages in scenarios where traditional implicit-differentiation-based methods fall short, such as models that have not fully converged or those trained in multiple stages. Our empirical evidence suggests that SOURCE is a valuable TDA tool in various scenarios.

## 2 Background

Consider a finite training dataset $\mathcal{D} := \{z_i\}_{i=1}^{N}$. We assume that the model parameters $\boldsymbol{\theta} \in \mathbb{R}^D$ are optimized with a gradient-based iterative optimizer to minimize the empirical risk on this dataset:

$$\mathcal{J}(\boldsymbol{\theta}, \mathcal{D}) := \frac{1}{N} \sum_{i=1}^{N} \mathcal{L}(z_i, \boldsymbol{\theta}), \tag{1}$$

where $\mathcal{L}$ is the (twice-differentiable) loss function. We use the notation $\boldsymbol{\theta}^\star(\mathcal{S})$ to denote the optimal solution obtained when the model is trained on a specific subset of the dataset $\mathcal{S} \subseteq \mathcal{D}$, and $\boldsymbol{\theta}^\star$ to denote the optimal solution on the full dataset. In practice, it is common to employ parameters $\boldsymbol{\theta}^s$ that approximately minimize the empirical risk (*e.g.*, the result of running an optimization algorithm for $T$ iterations), as obtaining the exact optimal solution for neural networks can be challenging and may lead to overfitting [7]. When necessary, we use the notation $\boldsymbol{\theta}^s(\mathcal{S}; \boldsymbol{\lambda}, \xi)$ to indicate the final parameters obtained by training with the dataset $\mathcal{S}$, along with hyperparameters $\boldsymbol{\lambda}$ (*e.g.*, learning rate and number of epochs) and random choices $\xi$ (*e.g.*, parameter initialization and mini-batch order).

### 2.1 Training Data Attribution

TDA aims to explain model behavior on a query data point $z_q$ (*e.g.*, test example) by referencing data points used to fit the model. The model behavior is typically quantified using a measurement $f(z_q, \boldsymbol{\theta})$, selected based on metrics relevant to the analysis, such as loss, margin, or log probability. Given hyperparameters $\boldsymbol{\lambda}$ and a training data point $z_m \in \mathcal{D}$, an attribution method $\tau(z_q, z_m, \mathcal{D}; \boldsymbol{\lambda})$ assigns

a score to a training data point, indicating its *importance* in influencing the expected measurable quantity $\mathbb{E}_\xi\left[f(z_q, \theta^s(\mathcal{D}; \lambda, \xi))\right]$, where the expectation is taken over the randomness in the training process. In cases where an optimal solution to Equation (1) exists, is unique, and can be precisely computed, and TDA is performed on this optimal solution, the TDA method is simply written as $\tau(z_q, z_m, \mathcal{D})$.

One idealized TDA method is *leave-one-out* (LOO) retraining [88], which assesses a data point's importance through counterfactual analysis. Assuming the above optimality condition is satisfied, for a chosen query data point $z_q$ and a training data point $z_m \in \mathcal{D}$, the LOO score can be formulated as:

$$\tau_{\text{LOO}}(z_q, z_m, \mathcal{D}) := f(z_q, \theta^\star(\mathcal{D} \setminus \{z_m\})) - f(z_q, \theta^\star). \tag{2}$$

When the measurement is defined as the loss, a higher absolute LOO score signifies a more substantial change in the query loss when the data point $z_m$ is excluded from the training dataset, particularly when the model parameters are optimized for convergence.

## 2.2 Influence Functions

Influence functions estimate the change in optimal parameters resulting from an infinitesimal perturbation in the weight of a training example $z_m \in \mathcal{D}$. Assuming that an optimal solution to Equation (1) exists and is unique for various values of the data point's weight $\epsilon \in [-1, 1]$, the relationship between this weight and the optimal parameters is captured through the *response function*:

$$r(\epsilon) := \arg\min_{\theta} \mathcal{J}(\theta, \mathcal{D}) + \frac{\epsilon}{N}\mathcal{L}(z_m, \theta). \tag{3}$$

Influence functions approximate Equation (3) using the first-order Taylor expansion around $\epsilon = 0$:

$$r(\epsilon) \approx r(0) + \frac{\mathrm{d}r}{\mathrm{d}\epsilon}\Big|_{\epsilon=0} \cdot \epsilon = \theta^\star - \mathbf{H}^{-1}\nabla_\theta \mathcal{L}(z_m, \theta^\star)\epsilon, \tag{4}$$

where the Jacobian of the response function $\mathrm{d}r/\mathrm{d}\epsilon|_{\epsilon=0}$ is obtained using the Implicit Function Theorem [52] and $\mathbf{H} := \nabla_\theta^2 \mathcal{J}(\theta^\star, \mathcal{D})$ represents the Hessian of the cost function at the optimal solution. The change in the optimal parameters due to the removal of $z_m$ can be approximated by setting $\epsilon = -1$:

$$\theta^\star(\mathcal{D} \setminus \{z_m\}) - \theta^\star \approx \frac{1}{N}\mathbf{H}^{-1}\nabla_\theta \mathcal{L}(z_m, \theta^\star). \tag{5}$$

By applying the chain rule of derivatives, influence functions estimate the change in a measurable quantity for a query example $z_q$ as:

$$\tau_{\text{IF}}(z_q, z_m, \mathcal{D}) := \nabla_\theta f(z_q, \theta^\star)^\top \mathbf{H}^{-1}\nabla_\theta \mathcal{L}(z_m, \theta^\star). \tag{6}$$

We refer readers to Koh and Liang [49] for detailed derivations and discussions of influence functions. As observed in Equation (6), influence functions provide algorithms that only depend on the optimal parameters $\theta^\star$ rather than intermediate checkpoints. However, when applied to neural networks, the connection to the counterfactual prediction is tenuous due to the unrealistic assumptions that the optimal solution exists, is unique, and can be found [6, 3, 77]. In practice, the gradients and Hessian are computed using the final parameters $\theta^s$ from a single training run instead of the optimal solution.

# 3 Methods

In this section, we introduce SOURCE, a gradient-based TDA technique combining the advantages of implicit and unrolled differentiation. We motivate our approach from the unrolling perspective and, after introducing suitable approximations, arrive at an influence-function-like algorithm. Finally, we describe a practical instantiation of SOURCE by approximating the Hessian with the Eigenvalue-corrected Kronecker-Factored Approximate Curvature (EK-FAC) [18] parameterization.

## 3.1 Motivation: Unrolling for Training Data Attribution

Consider optimizing the model parameters using SGD with a fixed batch size $B$, starting from the initial parameters $\theta_0$.[1] The update rule at each iteration is expressed as follows:

$$\theta_{k+1} \leftarrow \theta_k - \frac{\eta_k}{B}\sum_{i=1}^{B}\nabla_\theta \mathcal{L}(z_{ki}, \theta_k), \tag{7}$$

---

[1]For an extension to preconditioned gradient updates, see Appendix C.

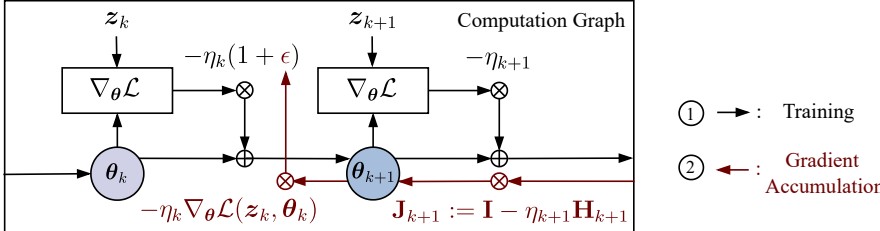

Figure 1: A simplified illustration of unrolled differentiation in SGD with a batch size of 1 and a data point of interest $z_m$ appearing at iteration $k$. Unrolling backpropagates through the optimization steps from $\theta_T$ to compute the total derivative with respect to $\epsilon$.

where $\eta_k$ denotes the learning rate for iteration $k$, $z_{ki}$ is the $i$-th data point in $\mathcal{B}_k$, where $\mathcal{B}_k$ denotes a mini-batch of examples drawn randomly with replacement from the training dataset $\mathcal{D}$, and $T$ denotes the total number of iterations. We aim to understand the effect of removing a training data point $z_m \in \mathcal{D}$ on the terminal model parameters $\theta_T$. To this end, we parameterize the weight of $z_m$ as $1 + \epsilon$, where $\epsilon = 0$ corresponds to the original training run and $\epsilon = -1$ represents the removal of a data point. This parameterization results in the following update rule:

$$\theta_{k+1}(\epsilon) \leftarrow \theta_k(\epsilon) - \frac{\eta_k}{B} \sum_{i=1}^{B} (1 + \delta_{ki}\epsilon) \nabla_{\theta}\mathcal{L}(z_{ki}, \theta_k(\epsilon)), \tag{8}$$

where $\delta_{ki} := \mathbb{1}[z_{ki} = z_m]$ is the indicator function for having selected $z_m$. The dependence of $\theta$ on $\epsilon$ will usually be suppressed for brevity. Similarly to other gradient-based TDA methods, such as influence functions, we approximate the change in the terminal parameters due to the data removal $\theta_T(-1) - \theta_T(0)$ with its first-order Taylor approximation $\mathrm{d}\theta_T/\mathrm{d}\epsilon$ (the notation $|_{\epsilon=0}$ is suppressed as it will always be evaluated at $\epsilon = 0$). Let $\delta_k$ denote the number of times $z_m$ is chosen in batch $\mathcal{B}_k$. By chain rule, the contribution of iteration $k$ to the total derivative $\mathrm{d}\theta_T/\mathrm{d}\epsilon$ can be found by multiplying all the Jacobian matrices along the accumulation path, giving the value $-\frac{\eta_k}{B}\delta_k \mathbf{J}_{k+1:T}\mathbf{g}_k$, where:

$$\mathbf{J}_k := \frac{\mathrm{d}\theta_{k+1}}{\mathrm{d}\theta_k} = \mathbf{I} - \eta_k\mathbf{H}_k, \quad \mathbf{J}_{k:k'} := \frac{\mathrm{d}\theta_{k'}}{\mathrm{d}\theta_k} = \mathbf{J}_{k'-1}\cdots\mathbf{J}_{k+1}\mathbf{J}_k, \quad \mathbf{g}_k := \nabla_{\theta}\mathcal{L}(z_m, \theta_k). \tag{9}$$

Here, $\mathbf{H}_k := \frac{1}{B}\sum_{i=1}^{B} \nabla_{\theta}^2 \mathcal{L}(z_{ki}, \theta_k)$ is the mini-batch Hessian for iteration $k$ and we define $\mathbf{J}_{k:k} := \mathbf{I}$ for any $0 \leq k < T$ by convention. A simplified illustration of unrolling is shown in Figure 1.

In contrast to influence functions, unrolling does not assume uniqueness or convergence to the optimal solution. An illustrative comparison of the two approaches is shown in Figure 2. Exact influence functions differentiate the response function (Equation (4)), estimating the sensitivity of the optimal solution ($\star$) to downweighting a data point. By contrast, unrolling estimates the sensitivity of the *final* model parameters (at the end of training) to downweighting a data point; hence, it can account for details of the training process such as learning rate schedules, implicit bias of optimizers, or a data point's position during training. For instance, in our illustrative example, gradient descent optimization is stopped early, such that the optimizer makes much progress in the high curvature direction and little in the low curvature direc-

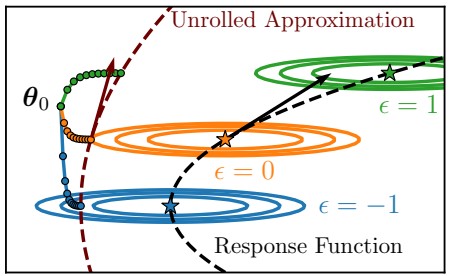

Figure 2: Illustrative comparision of influence functions and unrolling-based TDA. Each contour represents the cost function at different values of $\epsilon$, which controls the degree of downweighting a data point $z_m$.

tion. Unrolling-based TDA (but not implicit differentiation) accounts for this effect, resulting in a smaller influence along the low curvature direction.

The effect of removing $z_m$ on any single training trajectory may be noisy and idiosyncratic. For stability, we instead consider the expectation over training trajectories, where the selection of training examples in each batch (and all downstream quantities such as the iterates $\theta_k$) are treated as random variables.[2] We are interested in the average treatment effect $\mathbb{E}[\theta_T(-1) - \theta_T(0)]$, where the expectation is over the batch selection, and approximate this quantity with $-\mathbb{E}[\mathrm{d}\theta_T/\mathrm{d}\epsilon]$. The expected

---

[2]We assume a fixed initialization $\theta_0$ to break the symmetry.

total derivative can be expanded as a sum over all iterations, applying linearity of expectation:

$$\mathbb{E}\left[\frac{\mathrm{d}\boldsymbol{\theta}_T}{\mathrm{d}\epsilon}\right] = \mathbb{E}\left[-\sum_{k=0}^{T-1}\frac{\eta_k}{B}\delta_k\mathbf{J}_{k+1:T}\mathbf{g}_k\right] = -\sum_{k=0}^{T-1}\frac{\eta_k}{B}\mathbb{E}\left[\delta_k\mathbf{J}_{k+1:T}\mathbf{g}_k\right]. \tag{10}$$

In principle, we could compute a Monte Carlo estimate of this expectation by averaging many training trajectories. For each trajectory, $\mathrm{d}\boldsymbol{\theta}_T/\mathrm{d}\epsilon$ can be evaluated using reverse accumulation (*i.e.*, backpropagation) on the computation graph. However, this approach is prohibitively expensive as it requires storing all intermediate variables for the backward pass. Furthermore, many Monte Carlo samples may be required to achieve accurate estimates.

## 3.2 Segmenting the Training Trajectory

To derive a more efficient algorithm for approximating expected total derivative $\mathbb{E}\left[\mathrm{d}\boldsymbol{\theta}_T/\mathrm{d}\epsilon\right]$, we now partition the training procedure into $L$ segments and approximate the reverse accumulation computations for each segment with statistical summaries thereof (instead of storing all intermediate variables). Our motivations for segmenting the training procedure are twofold. First, the training procedure may explicitly include multiple stages with distinct objectives or datasets, as in continual learning or foundation models. Second, the Hessians and gradients are likely to evolve significantly over training, and segmenting the training allows us to approximate their distributions as stationary within a segment (rather than over the entire training run).

We index the segments as $\ell = 1, \ldots, L$, with segment boundaries denoted as $T_\ell$. By convention, $T_L := T$ and $T_0 := 0$ denote the end of training and beginning of training, respectively, and $K_\ell := T_\ell - T_{\ell-1}$ denotes the total number of iterations within a segment. Conceptually, we can compute $\mathrm{d}\boldsymbol{\theta}_T/\mathrm{d}\epsilon$ using reverse accumulation over a coarse-grained computation graph represented in terms of segments rather than individual iterations. The Jacobian associated with each segment is denoted as $\mathbf{S}_\ell := \mathbf{J}_{T_{\ell-1}:T_\ell}$. To approximate the expected total derivative $\mathbb{E}\left[\mathrm{d}\boldsymbol{\theta}_T/\mathrm{d}\epsilon\right]$, we first rewrite Equation (10) using the segment notation introduced. We then approximate the Jacobians of different segments as statistically independent (see discussion below):

$$\mathbb{E}\left[\frac{\mathrm{d}\boldsymbol{\theta}_T}{\mathrm{d}\epsilon}\right] = -\mathbb{E}\left[\sum_{\ell=1}^{L}\left(\prod_{\ell'=L}^{\ell+1}\mathbf{S}_{\ell'}\right)\underbrace{\sum_{k=T_{\ell-1}}^{T_\ell-1}\frac{\eta_k}{B}\delta_k\mathbf{J}_{k+1:T_\ell}\mathbf{g}_k}_{:=\mathbf{r}_\ell}\right] \approx -\sum_{\ell=1}^{L}\left(\prod_{\ell'=L}^{\ell+1}\mathbb{E}[\mathbf{S}_{\ell'}]\right)\mathbb{E}\left[\mathbf{r}_\ell\right], \tag{11}$$

where $\approx$ uses our independence approximation to push the expectations inward. Note that our product notation $\prod_{\ell'=L}^{\ell+1}$ takes $\ell'$ in decreasing order from $L$ down to $\ell+1$.

To obtain tractable approximations for $\mathbb{E}[\mathbf{S}_\ell]$ and $\mathbb{E}[\mathbf{r}_\ell]$, we approximate the Hessian and gradients distributions as stationary within each segment. This implies that the Hessians within a segment share a common mean $\bar{\mathbf{H}}_\ell := \mathbb{E}[\mathbf{H}_k]$ for $T_{\ell-1} \leq k < T_\ell$. Analogously, the gradients within a segment share a common mean $\bar{\mathbf{g}}_\ell := \mathbb{E}[\mathbf{g}_k]$. Moreover, we approximate the step sizes within each segment with their mean $\bar{\eta}_\ell$. If these stationarity approximations are too inaccurate (*e.g.*, $\mathbb{E}[\mathbf{H}_k]$ and/or $\mathbb{E}[\mathbf{g}_k]$ change rapidly throughout the segment), one can improve the fidelity by carving the training trajectory into a larger number of segments, at the expense of increased computational and memory requirements. Finally, we approximate the Hessians and gradients in different time steps as statistically independent.[3]

**Approximation of $\mathbb{E}[\mathbf{S}_\ell]$.** We approximate $\mathbb{E}[\mathbf{S}_\ell]$ in Equation (11) as follows:

$$\mathbb{E}[\mathbf{S}_\ell] = \mathbb{E}[\mathbf{J}_{T_{\ell-1}:T_\ell}] \approx \left(\mathbf{I} - \bar{\eta}_\ell\bar{\mathbf{H}}_\ell\right)^{K_\ell} \approx \exp(-\bar{\eta}_\ell K_\ell \bar{\mathbf{H}}_\ell) := \bar{\mathbf{S}}_\ell, \tag{12}$$

where the first $\approx$ uses the stationary and independence approximations, and the second $\approx$ uses the definition of matrix exponential. One can gain an intuition for $\bar{\mathbf{S}}_\ell$ by observing that it is a matrix

---

[3]There are two sources of randomness in the gradient and Hessian at each step: the mini-batch sampling, and the optimization iterates (which, recall, we treat as random variables). Mini-batch sampling contributes to independent variability in different steps. However, autocorrelation of optimization iterates induces correlations between Hessians and gradients in different time steps. Our independence approximation amounts to neglecting these correlations.

function of $\bar{\mathbf{H}}_\ell$.[4] Let $\bar{\mathbf{H}}_\ell = \mathbf{Q}\boldsymbol{\Lambda}\mathbf{Q}^\top$ be the eigendecomposition of $\bar{\mathbf{H}}_\ell$ and let $\sigma_j$ be the $j$-th eigenvalue of $\bar{\mathbf{H}}_\ell$. The expression in Equation (12) can be seen as applying the function $F_{\mathbf{S}}(\sigma) \coloneqq \exp(-\bar{\eta}_\ell K_\ell \sigma)$ to each of the eigenvalues $\sigma$ of $\bar{\mathbf{H}}_\ell$. The value is close to zero in high-curvature directions, so the training procedure "forgets" the components of $\boldsymbol{\theta}$ which lie in these directions. However, information about $\boldsymbol{\theta}$ is retained throughout the $\ell$-th segment for low-curvature directions.

**Approximation of $\mathbb{E}[\mathbf{r}_\ell]$.** We further approximate $\mathbb{E}[\mathbf{r}_\ell]$ as follows:

$$\mathbb{E}[\mathbf{r}_\ell] = \mathbb{E}\left[\sum_{k=T_{\ell-1}}^{T_\ell-1} \frac{\eta_k}{B} \delta_k \mathbf{J}_{k+1:T_\ell} \mathbf{g}_k\right] \approx \frac{1}{N} \sum_{k=T_{\ell-1}}^{T_\ell-1} \bar{\eta}_\ell (\mathbf{I} - \bar{\eta}_\ell \bar{\mathbf{H}}_\ell)^{T_\ell-1-k} \bar{\mathbf{g}}_\ell \tag{13}$$

$$= \frac{1}{N}(\mathbf{I} - (\mathbf{I} - \bar{\eta}_\ell \bar{\mathbf{H}}_\ell)^{K_\ell})\bar{\mathbf{H}}_\ell^{-1}\bar{\mathbf{g}}_\ell \approx \frac{1}{N} \underbrace{(\mathbf{I} - \exp(-\bar{\eta}_\ell K_\ell \bar{\mathbf{H}}_\ell))\bar{\mathbf{H}}_\ell^{-1}}_{\coloneqq F_{\mathbf{r}}(\sigma)} \bar{\mathbf{g}}_\ell \coloneqq \bar{\mathbf{r}}_\ell, \tag{14}$$

where Equation (13) uses the stationary and independence approximations and $\mathbb{E}[\delta_k] = {}^B\!/\!_N$, and Equation (14) uses the finite series[5] and the definition of the matrix exponential. Because all the matrices commute, $\bar{\mathbf{r}}_\ell$ can also be written in terms of a matrix function, defined as $F_{\mathbf{r}}(\sigma) \coloneqq (1 - \exp(-\bar{\eta}_\ell K_\ell \sigma))/\sigma$. In high-curvature directions, this term approaches to ${}^1\!/\!_\sigma$, whereas in low-curvature directions, it approaches to $\bar{\eta}_\ell K_\ell$. The qualitative behavior of $F_{\mathbf{r}}$ can be captured with the function $F_{\text{inv}}(\sigma) \coloneqq 1/(\sigma + \lambda)$, where $\lambda = \bar{\eta}_\ell^{-1} K_\ell^{-1}$, as shown in Figure 6 (Appendix C). Applying this to $\bar{\mathbf{H}}_\ell$ results in approximating Equation (14) with the damped inverse-Hessian-vector product $(\bar{\mathbf{H}}_\ell + \lambda \mathbf{I})^{-1}\bar{\mathbf{g}}_\ell$. This is essentially the formula for influence functions, except that $\bar{\mathbf{H}}_\ell$ and $\bar{\mathbf{g}}_\ell$ represent the expected Hessian and gradient rather than the terminal one, and our analysis yields an explicit formula for the damping parameter $\lambda$. Hence, influence functions can be regarded approximately as a special case with only a single segment, so our damped unrolling analysis gives an alternative motivation for influence functions.

**Full Procedure.** We derived a closed-form term to approximate the expected total derivative:

$$\mathbb{E}\left[\frac{\mathrm{d}\boldsymbol{\theta}_T}{\mathrm{d}\epsilon}\right] \approx -\frac{1}{N} \sum_{\ell=1}^{L} \left(\prod_{\ell'=L}^{\ell+1} \bar{\mathbf{S}}_{\ell'}\right) \bar{\mathbf{r}}_\ell, \tag{15}$$

where $\bar{\mathbf{S}}_\ell$ and $\bar{\mathbf{r}}_\ell$ are obtained with Equation (12) and Equation (14), respectively. We term our algorithm SOURCE (**S**egmented stati**O**nary **U**n**R**olling for **C**ounterfactual **E**stimation) and refer readers to Figure 3 for a visual illustration. Similarly to unrolling, SOURCE can incorporate fine-grained information about optimization trajectories into the analysis. For instance, SOURCE can support TDA for non-converged models, accounting for the total number of iterations $T$ the model was trained with. It can also support TDA for multi-stage training pipelines: when the model was sequentially trained with two datasets $\mathcal{D}_1$ and $\mathcal{D}_2$, SOURCE can compute the contribution of a data point $\boldsymbol{z}_m \in \mathcal{D}_1$ that appeared in the first segment by partitioning the training trajectory into two segments and computing the expected total derivative at the first segment with $-\frac{1}{N_1}\bar{\mathbf{S}}_2\bar{\mathbf{r}}_1$, where $N_1$ is the size of the first training dataset.

Given terminal parameters $\boldsymbol{\theta}_T$ from a single training run and a query data point $\boldsymbol{z}_q$, SOURCE approximates the change in the measurable quantity due to the removal of a training data point $\boldsymbol{z}_m$ as:

$$\tau_{\text{SOURCE}}(\boldsymbol{z}_q, \boldsymbol{z}_m, \mathcal{D}; \boldsymbol{\lambda}) \coloneqq \nabla_{\boldsymbol{\theta}} f(\boldsymbol{z}_q, \boldsymbol{\theta}_T)^\top \left(\sum_{\ell=1}^{L} \left(\prod_{\ell'=L}^{\ell+1} \bar{\mathbf{S}}_{\ell'}\right) \bar{\mathbf{r}}_\ell\right). \tag{16}$$

Unlike the single-training-run estimator for unrolling-based approaches, SOURCE does not require access to the exact location where the data point $\boldsymbol{z}_m$ was used during training, as it estimates the averaged effect of removing a data point within a given segment. To further account for other sources of randomness, such as model initialization, the multiple-training-run estimator for SOURCE averages the final scores in Equation (16) obtained for each training run with different random choices.

---

[4]Given a scalar function $F$ and a square matrix $\mathbf{M}$ diagonalizable as $\mathbf{M} = \mathbf{P}\mathbf{D}\mathbf{P}^{-1}$, the matrix function is defined as $F(\mathbf{M}) = \mathbf{P}F(\mathbf{D})\mathbf{P}^{-1}$, where $F(\mathbf{D})$ applies $F$ to each diagonal entry of $\mathbf{D}$.

[5]For a symmetric square matrix $\mathbf{M}$, we have $\sum_{i=0}^{T-1} \mathbf{M}^i = (\mathbf{I} - \mathbf{M}^T)(\mathbf{I} - \mathbf{M})^{-1}$. When $\mathbf{I} - \mathbf{M}$ is singular, we can replace $(\mathbf{I} - \mathbf{M})^{-1}$ with the pseudoinverse $(\mathbf{I} - \mathbf{M})^+$.

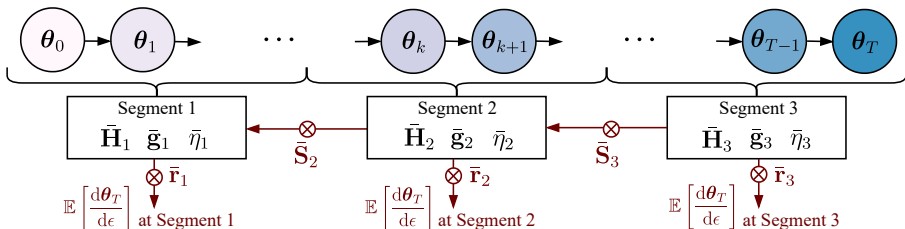

Figure 3: A simplified illustration of SOURCE with 3 segments ($L = 3$). SOURCE divides the training trajectory into one or more segments and approximate the gradient $\bar{\mathbf{g}}_\ell$ and Hessian $\bar{\mathbf{H}}_\ell$ distributions and learning rate $\bar{\eta}_\ell$ as stationary within each segment $\ell$ to approximate unrolling. SOURCE does not require storing the entire intermediate variables throughout training. Instead, it requires a handful of checkpoints throughout training to approximate the means of the Hessians and gradients.

## 3.3 Practical Algorithm for SOURCE

We now describe an instantiation of SOURCE which is practical to implement. Given the $C$ model checkpoints saved during training, SOURCE begins by organizing them into $L$ distinct segments. These segments may represent explicit stages in training (*e.g.*, continual learning) or account for the change in Hessian and gradient throughout training. Within each segment $\ell$, SOURCE estimates the stationary Hessian $\bar{\mathbf{H}}_\ell$ and gradient $\bar{\mathbf{g}}_\ell$ by averaging the Hessian and gradient across all checkpoints in the segment. SOURCE further estimates $\bar{\eta}_\ell$ by averaging the learning rates used within a segment.

However, computing Equation (15) has two practical bottlenecks for neural networks: computation of the Hessian and its matrix exponential. We fit a parametric approximation to the Hessian using EK-FAC [18]. EK-FAC parameterization is convenient for SOURCE as the approximate Hessian has an explicit eigendecomposition, which enables efficient computation of $\bar{\mathbf{S}}_\ell$ and $\bar{\mathbf{r}}_\ell$ by applying appropriate matrix functions to the eigenvalues. Note that EK-FAC approximates the Hessian with the Gauss-Newton Hessian (GNH) [63]. Unlike the Hessian, the GNH is guaranteed to be positive semi-definite, as long as the loss function is convex in the model outputs [62]. The GNH approximation within EK-FAC is also advantageous for SOURCE as it can avoid numerical instability in computing Equation (15), especially when the Hessian has negative eigenvalues. The implementation details are provided in Appendix D.

Compared to influence functions with the same EK-FAC approximation [24], SOURCE requires computing the EK-FAC factors and training gradients for each model checkpoint when performing TDA on all segments. Hence, SOURCE is $C$ times more computationally expensive, where $C$ is the number of checkpoints. In Appendix F.2, we introduce a more computationally efficient version of SOURCE, where we average the parameters within a segment instead of averaging Hessians and gradients. This variant of SOURCE is $L$ times more computationally expensive than influence functions, as the EK-FAC factors and gradients only need to be computed once for each segment.

While we described one instantiation of SOURCE with the EK-FAC approximation, we note that SOURCE can be integrated with other techniques used for approximating implicit-differentiation-based TDA methods, such as TRAK [70], DATAINF [55], and LOGRA [11]. For example, with TRAK, we can use random projection [43] and efficiently compute the averaged Hessian and gradients in the lower-dimensional space. TRAK can be advantageous over the EK-FAC approximation when there are a large number of query data points, as it caches the compressed training gradients in memory, avoiding the need to recompute them for each query.

## 4 Related Works

Modern TDA techniques for neural networks can be broadly categorized into three main groups: sampling-based, representation-based, and gradient-based. For a comprehensive overview of TDA, including practical applications, we refer the reader to Hammoudeh and Lowd [27] and Mucsányi et al. [66]. Sampling-based (or retraining-based) approaches, such as Shapley-value estimators [78, 21, 39, 54, 87], DOWNSAMPLING [17, 96], DATAMODELS [35], and DATA BANZHAF [5, 86], approximate counterfactuals by repeatedly retraining models on different data subsets. Although effective, these methods are often impractical for modern neural networks due to the significant computational cost of repeated model retraining.

Representation-based techniques evaluate the relevance between a training and query data point by examining the similarity in their representation space (*e.g.*, the output of the last hidden layer) [9, 30]. These techniques offer computational advantages compared to other attribution methods, as they only require forward passes through the trained network. Rajani et al. [74] further improves efficiency by caching all hidden representations of the training dataset and using approximate nearest neighbor search [42]. Past works have also proposed model-agnostic TDA approaches, such as computing the similarity between query and training sequences with BM25 [75] for language models [1, 56] or with an embedding vector obtained from a separate pre-trained self-supervised model for image classification tasks [79]. However, representation-based and input-similarity-based techniques lack a connection to the counterfactual and do not provide a notion of negatively influential data points.

Two main strategies for gradient-based TDA are implicit differentiation and unrolling. To the best of our knowledge, the largest model to which exact unrolling has been applied is a 300 thousand parameter model [31]. Our experiments in Section 5 cover TDA for models ranging from 560 thousand parameters (MNIST & MLP) to 120 million parameters (WikiText-2 & GPT-2). SGD-INFLUENCE [31] also considers applying unrolling to only the last epoch for large-scale models. However, this limits its applicability in analyzing the effect of removing a data point at the beginning of training or analyzing multi-stage training processes. In contrast, HYDRA [10] approximates the mini-batch Hessian $\mathbf{H}_k$ in Equation (10) as zero when computing the total derivatives, avoiding the need to compute Hessian-vector products (HVPs) for each optimization step. However, in Appendix F.1, we empirically observe that an accurate approximation of the Hessian is important to achieve good TDA performance. Both approaches require storing a large number of optimization variables during training. Relatedly, Nickl et al. [68] use local perturbation methods [37] to approximate the data point's sensitivity to the training trajectory.

Apart from implicit-differentiation-based and unrolling-based approaches, TRACIN [72] is another prominent gradient-based TDA technique, which estimates the importance of a training data point by approximating the total change in the query's measurable quantity with the gradient update from this data point throughout training. Similarly to SOURCE, the practical version of TRACIN (TRACINCP) leverages intermediate checkpoints saved during training. While TRACINCP is straightforward to implement as it does not involve approximation of the Hessians, its connection to the counterfactual is unclear [27, 77]. However, past works have shown its strengths in downstream tasks, such as mislabeled data detection [72] and curating fine-tuning data [90].

## 5  Experiments

Our experiments investigate two key questions: (1) How does SOURCE compare to existing TDA techniques, as measured by the linear datamodeling score (LDS) [70] and through subset removal counterfactual evaluation [33, 93, 35, 97, 70, 8, 79, 19]? (2) Can SOURCE support data attribution in situations where implicit-differentiation-based approaches struggle, particularly with models that have not converged or have been trained in multiple stages with different objectives or datasets?

We compare SOURCE against existing TDA techniques: representation similarity (REPSIM) [9, 30], TRACIN [72], TRAK [70], and influence functions (IF) with the EK-FAC approximation [24]. For consistency with Park et al. [70], the measurement $f$ is defined as the margin for classification tasks and the absolute error for regression tasks. Our evaluations are conducted under two separate settings. First is a single model setup, where TDA techniques use model checkpoints from a single training run. Unless specified otherwise, REPSIM, TRAK, and IF are computed at the final training checkpoint, and TRACIN and SOURCE use at most 6 intermediate checkpoints saved throughout training. In the second setting, TDA techniques use checkpoints from 10 distinct models, each trained with varying sources of randomness. Past works have shown that ensembling attribution scores across models can improve TDA performance [70, 67]. For all TDA techniques, including SOURCE, we simply average the final attribution scores from distinctly trained models with the full dataset, except for TRAK, which uses its custom ensembling procedures with models each trained on sampled 50% of the original dataset.

Our experiments consider diverse machine learning tasks, including: (a) regression using datasets from the UCI Repository [45], (b) image classification with datasets such as MNIST [57], FashionMNIST [91], CIFAR-10 [53], RotatedMNIST [20], and PACS [58], and (c) text classification using the GLUE benchmark [85]. Our tasks can be categorized into three groups:

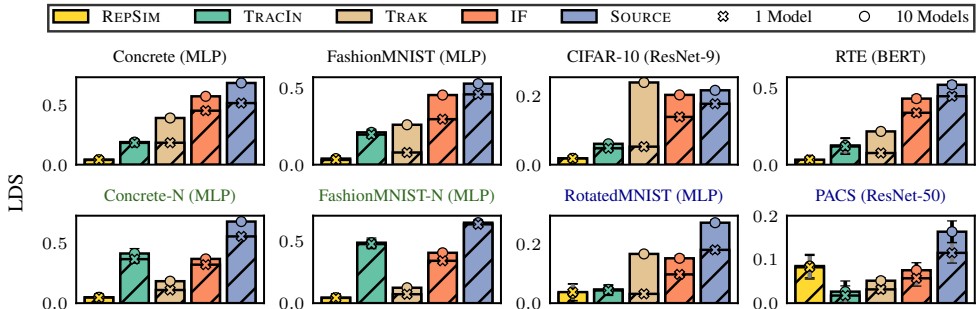

Figure 4: LDS at $\alpha = 0.5$ for SOURCE and baseline techniques on regression, image classification, and text classification tasks. The error bars represent $95\%$ bootstrap confidence intervals (Appendix E.2).

**Concrete, FashionMNIST, CIFAR-10, & RTE.** Models fully trained using a fixed dataset $\mathcal{D}$, where implicit-differentiation-based methods are expected to perform similarly to unrolling-based methods. We use 6 intermediate checkpoints throughout training for TRACIN and SOURCE. SOURCE use 3 segments ($L = 3$) equally partitioned at the early, middle, and late stages of training to account for the changes in distributions of Hessian and gradients during training.

**Concrete-N & FashionMNIST-N.** Non-converged models trained with a smaller number of update steps. This is a challenging setup for implicit-differentiation-based methods, such as TRAK and IF, as they inherently assume that TDA is performed on the optimal solution. We use versions of the Concrete and FashionMNIST datasets that have been modified, either by corrupting target values or relabeling $30\%$ of the data points. Then, we train the models for only 3 epochs to avoid overfitting. We use 3 checkpoints (at the end of each epoch) for TRACIN and SOURCE ($L = 3$).

**RotatedMNIST & PACS.** Models initially trained with a dataset $\mathcal{D}_1$, and subsequently trained with another dataset $\mathcal{D}_2$ (a common setup in continual learning). We use test examples from $\mathcal{D}_2$ for query data points and attribute the final model's behavior to the first dataset. Since implicit-differentiation-based methods do not provide any way to separate multiple stages of training, for TRAK and IF, we simply combine the data from both stages into a larger dataset for TDA. We use two segments for SOURCE, partitioned at different stages, and perform TDA only for the first segment. Our experiments use the RotatedMNIST and PACS datasets, both containing multiple data distributions. We select one of these domains for the second training stage, while the remaining ones are used in the first stage.

The detailed description of the experimental setup is provided in Appendix E. Additional results, including comparisons on additional tasks and with additional baselines, further analysis on linear models, and visualizations of the top influential images obtained by each TDA technique, are shown in Appendix F.

## 5.1 TDA Evaluations with Linear Datamodeling Score (LDS)

We evaluate TDA techniques using the linear datamodeling score (LDS) from Park et al. [70]. To compute LDS, we first generate $M$ random subsets $\{\mathcal{S}_j\}_{j=1}^{M}$ from the training dataset, each containing $\lceil \alpha N \rceil$ data points for some $\alpha \in (0, 1)$. Given a query data point $z_q$ and hyperparameters $\boldsymbol{\lambda}$ used to train the original model, the expected measurable quantity for each data subset $\mathbb{E}_{\xi}[f(z_q, \boldsymbol{\theta}^s(\mathcal{S}_j; \boldsymbol{\lambda}, \xi))]$ is estimated by retraining the model $R$ times under different random choices (which requires $MR$ model retrainings in total). The LDS measures the Spearman correlation [81] between the estimated quantities and the predictions made by the TDA method. Note that, although a TDA method in Section 2.1 assigns a score to each pair of a query and training data point, the inherently *additive* nature of most TDA techniques allows for the computation of a group prediction score for the data subset $\mathcal{S}$ by summing the individual scores attributed to each data point within this subset. The final LDS is obtained by averaging the scores across many (typically up to 2000) query data points. We use 100 data subsets ($M = 100$) and conduct a minimum of 5 retraining iterations ($R \geq 5$) for each subset. We refer readers to Appendix A for the detailed formulation and to Appendix E.2 for the practical procedures.

The LDS at $\alpha = 0.5$ for SOURCE and baseline TDA techniques are shown in Figure 4. SOURCE consistently outperforms all baseline methods in a single model setup, achieving high LDS. When aggregating TDA scores from multiple models, we observe a large improvement in the LDS, particu-

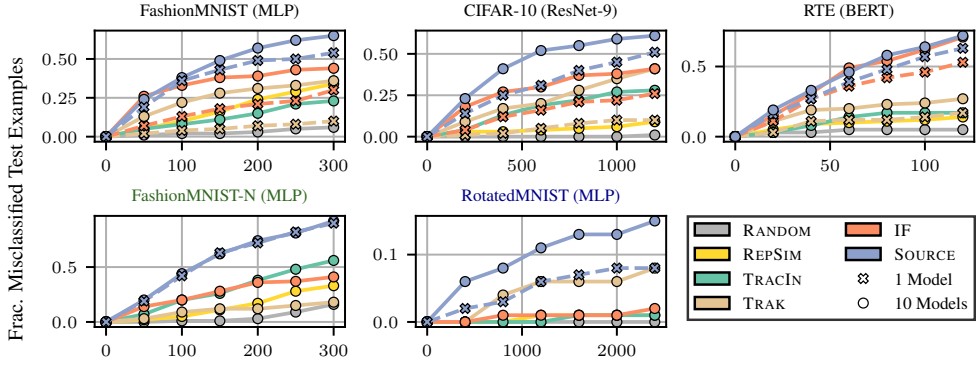

Figure 5: Subset removal counterfactual evaluation for SOURCE and baseline TDA techniques, where the top positively influential data points predicted by each TDA method are removed, and the model is retrained to examine if (previously correctly classified) test data point gets misclassified.

larly for TRAK, IF, and SOURCE. Our method achieves the highest LDS across all tasks, except for the CIFAR-10 classification task using ResNet-9. SOURCE especially performs strongly against other baseline techniques on settings that pose challenges to implicit-differentiation-based approaches (*e.g.*, non-converged models and models trained with multiple stages), and indeed, even the non-ensembled version of SOURCE typically outperforms the ensembled versions of the competing methods.

## 5.2 TDA Evaluations with Subset Removal

Subset removal counterfactual evaluation examines the change in model behavior before and after removing data points highly ranked by a TDA technique. For classification tasks, we consider 100 test data points that are correctly classified when trained with the full dataset (across all 5 random seeds) and, for each test data point $z_q$, examine if removing and retraining without the top-$k$ *positively* influential data points can cause misclassification on average (over 3 random seeds). By assessing the impact of removing influential training examples on the model's performance, counterfactual evaluation provides a direct measure of the effectiveness of TDA techniques in identifying data points that significantly contribute to the model's behavior. The detailed procedures are described in Appendix E.3. In Figure 5, we show the fraction of test examples (out of the selected 100 test points) that get misclassified on average after removing at most $k$ positively influential training examples identified by each TDA method. We observe that SOURCE better identifies the top influential data points causing misclassification than other baseline TDA techniques. The improvement is more substantial for settings that pose challenges to implicit-differentiation-based methods.

## 6 Conclusion

We introduced SOURCE (**S**egmented stati**O**nary **U**n**R**olling for **C**ounterfactual **E**stimation), a novel TDA technique that combines the strengths of implicit-differentiation-based and unrolling-based techniques. SOURCE approximates unrolled differentiation by partitioning the training trajectory into one or more segments and approximating the gradients and Hessians as stationary within each segment, yielding an influence-function-like estimation algorithm. We showed one instantiation of SOURCE by approximating the Hessian with the EK-FAC parameterization. On a diverse task set, we demonstrated SOURCE's effectiveness compared to existing data attribution techniques, especially when the network has not converged or has been trained with multiple stages.

## Acknowledgements

The authors would like to thank Jenny Bao, Rob Brekelmans, Sang Keun Choe, Lev McKinney, Andrew Wang, and Arielle Zhang for their helpful feedback on the manuscript. Resources used in preparing this research were provided, in part, by the Province of Ontario, the Government of Canada through CIFAR, and companies sponsoring the Vector Institute: `www.vectorinstitute.ai/#partners`. JB was funded by OpenPhilanthropy and Good Ventures. RG acknowledges support from the Canada CIFAR AI Chairs program.

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

# A  Evaluation of TDA Techniques

Given the focus on counterfactual prediction in many TDA methods, LOO estimates, defined in Equation (2), are often considered a ground truth for evaluating these techniques. However, the computation of LOO scores in neural networks encounters several computational and conceptual challenges, as detailed in Appendix B. For a robust and standardized measure for evaluating TDA techniques, we instead use the linear datamodeling score (LDS) from Park et al. [70] as well as subset removal counterfactual evaluation [33, 93, 35, 97, 70, 8, 79, 19].

**Linear Datamodeling Score (LDS).**   A TDA method $\tau$, as detailed in Section 2.1, assigns a score to each pair of a query and training data point. The inherently *additive* nature of most TDA techniques allows for the computation of a group attribution score for a specific training data subset $\mathcal{S} \subset \mathcal{D}$. The importance of $\mathcal{S}$ on the measurable quantity $f$ is estimated by summing the individual scores attributed to each data point within this subset. The group attribution is expressed as follows:

$$g_\tau(\boldsymbol{z}_q, \mathcal{S}, \mathcal{D}; \boldsymbol{\lambda}) := \sum_{\boldsymbol{z} \in \mathcal{S}} \tau(\boldsymbol{z}_q, \boldsymbol{z}, \mathcal{D}; \boldsymbol{\lambda}). \tag{17}$$

Consider $M$ random subsets $\{\mathcal{S}_j\}_{j=1}^M$ from the training dataset, each containing $\lceil \alpha N \rceil$ data points for some $\alpha \in (0, 1)$. Given a hyperparameter configuration $\boldsymbol{\lambda}$ to train the model, the LDS for a query point $\boldsymbol{z}_q$ is defined as:

$$\text{LDS}_\alpha(\boldsymbol{z}_q, \tau) := \boldsymbol{\rho}\left(\{\mathbb{E}_\xi\left[f(\boldsymbol{z}_q, \boldsymbol{\theta}^s(\mathcal{S}_j; \boldsymbol{\lambda}, \xi))\right] : j \in [M]\}, \{g_\tau(\boldsymbol{z}_q, \mathcal{S}_j, \mathcal{D}; \boldsymbol{\lambda}) : j \in [M]\}\right), \tag{18}$$

where $\boldsymbol{\rho}$ represents the Spearman correlation [81]. This expected measurable quantity is approximated by retraining the network $R$ times under different random choices. The final LDS is obtained by averaging the scores across many (typically up to 2000) query data points. In our experiments, we use 100 data subsets ($M = 100$) and conduct a maximum of 100 retraining iterations ($R \in \{5, 10, 20, 100\}$) for each subset to compute the LDS.

**Subset Removal Counterfactual Evaluation.**   Subset removal counterfactual evaluation examines the change in model behavior before and after removing data points that are highly ranked by an attribution technique. For classification tasks, we consider 100 test data points that are correctly classified when trained with the full dataset and, for each test data point, examine if removing and retraining without the top-$k$ *positively* influential data points can cause misclassification on average (trained under different random choices).[6] By assessing the impact of removing influential data points on the model's performance, counterfactual evaluation provides a direct measure of the effectiveness of TDA techniques in identifying data points that significantly contribute to the model's behavior.

**Downstream Task Evaluation.**   TDA techniques have also been evaluated on their performance on downstream tasks, such as mislabeled data detection [46, 72, 47], class detection [30, 55], finding hallucinations in the training dataset [56], and retrieving factual knowledge from the training dataset [1]. These tasks can offer additional insights into the effectiveness and applicability of data attribution methods in practical scenarios. However, the connections between these tasks and counterfactual prediction are often unclear [44, 70], and it is uncertain whether algorithmic improvements in counterfactual prediction will directly result in improved performance on these downstream tasks.

# B  Limitations of Leave-One-Out Estimates

The computation of leave-one-out (LOO) scores in Equation (2) presents several computational and conceptual challenges for neural networks. Firstly, calculating the LOO score for all training data points requires retraining the model $N$ times, where $N$ is the size of the training dataset. This process can be prohibitively expensive for large datasets and network architectures.

Moreover, the formulation of LOO assumes that an optimal solution to Equation (1) exists, is unique, and can be precisely computed, and that TDA is performed on this optimal solution. However, within the context of neural networks, these assumptions often do not hold, leading to ambiguities in the

---

[6]The literature also uses terms such as *helpful* [49], *proponent* [72], and *excitatory* [92] to describe positively influential training data points.

computation of LOO estimates. Previous works have investigated various LOO variants as a means to establish counterfactual ground truths [49, 6, 44, 40, 3, 14, 67]. For example, Koh and Liang [49] and Basu et al. [6] formulated the LOO ground truth by training the network for an additional number of steps from the final parameters $\boldsymbol{\theta}^s$ without a specific training data point. However, as noted by Bae et al. [3], these estimates may reflect the effect of training the network for additional steps instead of retraining without a data point, especially when the network has not converged.

A more standardized extension of LOO for neural networks is the *expected leave-one-out* (ELOO) retraining [44], formulated as:

$$\tau_{\text{ELOO}}(\boldsymbol{z}_q, \boldsymbol{z}_m, \mathcal{D}; \boldsymbol{\lambda}) := \mathbb{E}_\xi \left[ f(\boldsymbol{z}_q, \boldsymbol{\theta}^s(\mathcal{D} \setminus \{\boldsymbol{z}_m\}; \boldsymbol{\lambda}, \xi)) \right] - \mathbb{E}_\xi \left[ f(\boldsymbol{z}_q, \boldsymbol{\theta}^s(\mathcal{D}; \boldsymbol{\lambda}, \xi)) \right], \quad (19)$$

where $\boldsymbol{\lambda}$ denotes the hyperparameters used to train the model, and the expectation is taken over the randomness in the training process (typically estimated by retraining the network $R$ times). Note that the ELOO can also be seen as the ground truth for the linear datamodeling score (LDS) (defined in Appendix A) with $\alpha = 1 - 1/N$. Past works have demonstrated the unreliability of ELOO estimates due to stochasticity in model training, such as model initialization and batch ordering [44, 14, 67]. Specifically, Nguyen et al. [67] observed that the noise from the stochasticity often overshadows the actual signal of removing a single data point. In Appendix F.3, we also observe that the LDS significantly drops at $\alpha = 1 - 1/N$, suggesting that the counterfactual ground truth for removing a single data point can be difficult to obtain, as it can be extremely noisy for most training data examples.

## C   SOURCE with Preconditioning Matrix

In Section 3.1, we motivated our proposed algorithm, SOURCE, for cases where the parameters are optimized using stochastic gradient descent (SGD). In this section, we present the formulation of SOURCE when preconditioned optimizers, such as RMSProp [82], Adam [48], and K-FAC [63], are used to train the model.

To investigate the impact of removing a training data point $\boldsymbol{z}_m \in \mathcal{D}$, we follow a similar derivation as in Section 3.1, but now considering the preconditioning matrix:

$$\boldsymbol{\theta}_{k+1}(\epsilon) \leftarrow \boldsymbol{\theta}_k(\epsilon) - \frac{\eta_k}{B} \mathbf{P}_k \left( \sum_{i=1}^{B} (1 + \delta_{ki}\epsilon) \nabla_{\boldsymbol{\theta}} \mathcal{L}(\boldsymbol{z}_{ki}, \boldsymbol{\theta}_k(\epsilon)) \right), \quad (20)$$

where $\mathbf{P}_k$ is a (positive definite) preconditioning matrix and $\delta_{ki} := \mathbb{1}[\boldsymbol{z}_{ki} = \boldsymbol{z}_m]$ is the indicator function for having selected $\boldsymbol{z}_m$.

By applying the chain rule of derivatives, the contribution of iteration $k$ to the total derivative can be found by multiplying all the Jacobian matrices along the backward accumulation path, giving the value $-\frac{\eta_k}{B} \mathbf{J}_{k+1:T} \mathbf{P}_k \mathbf{g}_k$, where we have:

$$\begin{aligned} \mathbf{J}_k &:= \frac{\mathrm{d}\boldsymbol{\theta}_{k+1}}{\mathrm{d}\boldsymbol{\theta}_k} = \mathbf{I} - \eta_k \mathbf{P}_k \mathbf{H}_k \\ \mathbf{J}_{k:k'} &:= \frac{\mathrm{d}\boldsymbol{\theta}_{k'}}{\mathrm{d}\boldsymbol{\theta}_k} = \mathbf{J}_{k'-1} \cdots \mathbf{J}_{k+1} \mathbf{J}_k \\ \mathbf{g}_k &:= \nabla_{\boldsymbol{\theta}} \mathcal{L}(\boldsymbol{z}_m, \boldsymbol{\theta}_k). \end{aligned} \quad (21)$$

Hence, by applying the linearity of expectation, the expected total derivative of the terminal parameters $\boldsymbol{\theta}_T$ with respect to the perturbation $\epsilon$ is expressed as:

$$\mathbb{E}\left[ \frac{\mathrm{d}\boldsymbol{\theta}_T}{\mathrm{d}\epsilon} \right] = -\sum_{k=0}^{T-1} \frac{\eta_k}{B} \mathbb{E}[\delta_k \mathbf{J}_{k+1:T} \mathbf{P}_k \mathbf{g}_k], \quad (22)$$

As discussed in Section 3.2, we group the training trajectories into multiple segments to approximate the expected total derivative for each segment with statistical summaries thereof. In addition to the approximations introduced in Section 3.2, we approximate preconditioning matrices as stationary within a segment and represent it as $\tilde{\mathbf{P}}_\ell := \mathbf{P}_k$ for $T_{\ell-1} \leq k < T_\ell$.

**Approximation of $\mathbb{E}[\mathbf{S}_\ell]$.** We approximate $\mathbb{E}[\mathbf{S}_\ell]$ in Equation (11) as follows:

$$\mathbb{E}[\mathbf{S}_\ell] = \mathbb{E}[\mathbf{J}_{T_{\ell-1}:T_\ell}] \approx \left(\mathbf{I} - \bar{\eta}_\ell\bar{\mathbf{P}}_\ell\bar{\mathbf{H}}_\ell\right)^{K_\ell} \approx \exp(-\bar{\eta}_\ell K_\ell \bar{\mathbf{P}}_\ell\bar{\mathbf{H}}_\ell)$$
$$= \tilde{\mathbf{P}}_\ell^{1/2}\exp(-\bar{\eta}_\ell K_\ell \bar{\mathbf{P}}_\ell^{1/2}\bar{\mathbf{H}}_\ell\bar{\mathbf{P}}_\ell^{1/2})\tilde{\mathbf{P}}_\ell^{-1/2} := \bar{\mathbf{S}}_\ell. \tag{23}$$

Note that the last line uses the properties of the matrix exponential.[7]

**Approximation of $\mathbb{E}[\mathbf{r}_\ell]$.** We further approximate $\mathbb{E}[\mathbf{r}_\ell]$ as follows:

$$\mathbb{E}[\mathbf{r}_\ell] = \mathbb{E}\left[\sum_{k=T_{\ell-1}}^{T_\ell-1} \frac{\eta_k}{B}\delta_k\mathbf{J}_{k+1:T_\ell}\mathbf{P}_k\mathbf{g}_k\right] \tag{24}$$

$$\approx \frac{1}{N}\sum_{k=T_{\ell-1}}^{T_\ell-1}\bar{\eta}_\ell(\mathbf{I} - \bar{\eta}_\ell\tilde{\mathbf{P}}_\ell\bar{\mathbf{H}}_\ell)^{T_\ell-1-k}\tilde{\mathbf{P}}_\ell\bar{\mathbf{g}}_\ell \tag{25}$$

$$= \frac{1}{N}(\mathbf{I} - (\mathbf{I} - \bar{\eta}_\ell\tilde{\mathbf{P}}_\ell\bar{\mathbf{H}}_\ell)^{K_\ell})\bar{\mathbf{H}}_\ell^{-1}\bar{\mathbf{g}}_\ell \tag{26}$$

$$\approx \frac{1}{N}(\mathbf{I} - \exp(-\bar{\eta}_\ell K_\ell\tilde{\mathbf{P}}_\ell\bar{\mathbf{H}}_\ell))\bar{\mathbf{H}}_\ell^{-1}\bar{\mathbf{g}}_\ell \tag{27}$$

$$= \frac{1}{N}\tilde{\mathbf{P}}_\ell^{1/2}\underbrace{(\mathbf{I} - \exp(-\bar{\eta}_\ell K_\ell\mathbf{M}_\ell))\mathbf{M}_\ell^{-1}}_{:=F_{\mathbf{r}}}\tilde{\mathbf{P}}_\ell^{1/2}\bar{\mathbf{g}}_\ell := \bar{\mathbf{r}}_\ell, \tag{28}$$

where we define $\mathbf{M}_\ell := \bar{\mathbf{P}}_\ell^{1/2}\bar{\mathbf{H}}_\ell\bar{\mathbf{P}}_\ell^{1/2}$ and the last line uses the properties of matrix exponential, as done in Equation (23). Similarly to our analysis presented in Section 3.2, we can represent $\bar{\mathbf{r}}_\ell$ with the matrix function of $\mathbf{M}_\ell$. Let $\mathbf{M}_\ell = \mathbf{Q}\mathbf{\Lambda}\mathbf{Q}^\top$ be the eigendecomposition of $\mathbf{M}_\ell$ and let $\sigma_j$ be the $j$-th eigenvalue of $\mathbf{M}_\ell$. The expression can be seen as applying the matrix function, defined as:

$$F_{\mathbf{r}}(\sigma) := \frac{1 - \exp\left(-\bar{\eta}_\ell K_\ell\sigma\right)}{\sigma}. \tag{29}$$

The qualitative behavior of $F_{\mathbf{r}}$ can be captured with the function $F_{\text{inv}}(\sigma) := 1/(\sigma + \lambda)$, where $\lambda = \bar{\eta}_\ell^{-1}K_\ell^{-1}$ (see Section 3.2 for details). Hence, one way to understand Equation (28) is by expressing it as the damped inverse Hessian-vector product (iHVP):

$$\bar{\mathbf{r}}_\ell \approx \frac{1}{N}\bar{\mathbf{P}}_\ell^{1/2}(\mathbf{M}_\ell + \lambda\mathbf{I})^{-1}\bar{\mathbf{P}}_\ell^{1/2}\bar{\mathbf{g}}_\ell \tag{30}$$

$$= \frac{1}{N}\bar{\mathbf{P}}_\ell^{1/2}(\bar{\mathbf{P}}_\ell^{1/2}\bar{\mathbf{H}}_\ell\bar{\mathbf{P}}_\ell^{1/2} + \lambda\mathbf{I})^{-1}\bar{\mathbf{P}}_\ell^{1/2}\bar{\mathbf{g}}_\ell \tag{31}$$

$$= \frac{1}{N}(\bar{\mathbf{H}}_\ell + \lambda\tilde{\mathbf{P}}_\ell^{-1})^{-1}\bar{\mathbf{g}}_\ell. \tag{32}$$

In a case where $\tilde{\mathbf{P}}_\ell$ is a diagonal matrix, Equation (32) can be seen as a special case for influence functions with a specific diagonal damping term $\lambda\tilde{\mathbf{P}}_\ell^{-1}$. Using the derived $\bar{\mathbf{S}}_\ell$ and $\bar{\mathbf{r}}_\ell$, we approximate the total expected derivative using Equation (15).

# D  Implementation Details

This section describes the Eigenvalue-corrected Kronecker-Factored Approximate Curvature (EK-FAC) [18] and how we computed SOURCE using this EK-FAC parameterization. The code for implementing SOURCE (as well as baseline techniques) will be provided at `https://github.com/pomonam/kronfluence`. For details on the EK-FAC approximation specific to influence functions, we refer readers to Grosse et al. [24].

---

[7]For a square matrix $\mathbf{M}$ and a square positive definite matrix $\mathbf{D}$, we have $\exp(\mathbf{M}) = \sum_{k=0}^\infty \frac{1}{k!}\mathbf{M}^k = \mathbf{D}^{1/2}\left[\sum_{k=0}^\infty \frac{1}{k!}\left(\mathbf{D}^{-1/2}\mathbf{M}\mathbf{D}^{1/2}\right)^k\right]\mathbf{D}^{-1/2} = \mathbf{D}^{1/2}\exp(\mathbf{D}^{-1/2}\mathbf{M}\mathbf{D}^{1/2})\mathbf{D}^{-1/2}$.

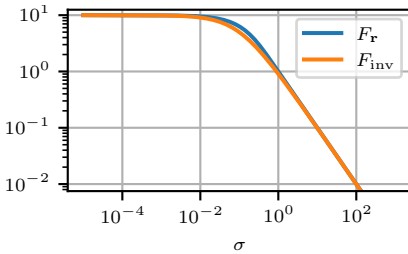

Figure 6: A demonstration of the match in qualitative behavior between $F_{\mathbf{r}}$ and $F_{\text{inv}}$, where we set $\bar{\eta}_\ell = 0.1$ and $K_\ell = 100$.

### D.1 Eigenvalue-corrected Kronecker-Factored Approximate Curvature (EK-FAC)

Kronecker-Factored Approximate Curvature (K-FAC) [63] and EK-FAC [18] introduce a parametric approximation to the Fisher information matrix (FIM) of a neural network, defined as:

$$\mathbf{F} := \mathbb{E}_{\boldsymbol{x} \sim p_{\text{data}}, \hat{\boldsymbol{y}} \sim P_{\hat{\boldsymbol{y}}|\boldsymbol{x}}(\boldsymbol{\theta})} \left[ \nabla_{\boldsymbol{\theta}} \log p(\hat{\boldsymbol{y}}|\boldsymbol{\theta}, \boldsymbol{x}) \nabla_{\boldsymbol{\theta}} \log p(\hat{\boldsymbol{y}}|\boldsymbol{\theta}, \boldsymbol{x})^\top \right], \tag{33}$$

where $p_{\text{data}}$ is the data distribution and $P_{\hat{\boldsymbol{y}}|\boldsymbol{x}}(\boldsymbol{\theta})$ is the model's output distribution. For many commonly used loss functions, such as softmax-cross-entropy and squared-error, the FIM is equivalent to the Gauss-Newton Hessian (GNH) [62], denoted as $\mathbf{G}$. The GNH can be seen as an approximation to the Hessian $\mathbf{H}$, where the network is linearized around the current parameters [22]. Different from the Hessian, the GNH is guaranteed to be positive semi-definite (PSD) when the loss function is convex with respect to the model output.

While K-FAC and EK-FAC were originally formulated for multilayer perceptrons (MLPs), they were later extended to other architectures, such as convolutional neural networks [23], recurrent neural networks [64], graph neural networks [36], or to be learnable by gradient-based optimizers [4]. We refer readers to Eschenhagen et al. [15] for a comprehensive overview. This section describes the EK-FAC formulation in the context of MLPs.

Consider a $l$-th layer of the network with input activations $\mathbf{a}_{l-1} \in \mathbb{R}^I$ and pre-activation output $\mathbf{s}_l \in \mathbb{R}^O$ such that $\mathbf{s}_l := \mathbf{W}_l \mathbf{a}_{l-1}$, where $\mathbf{W} \in \mathbb{R}^{O \times I}$ is the weight matrix (we drop the layer subscript to avoid clutter and ignore the bias term for simplicity). The pseudo-gradient (where the target is sampled from the model's output distribution; see Equation (33)) is given by $\mathcal{D}\mathbf{W} := \mathcal{D}\mathbf{s}\mathbf{a}^\top$. K-FAC makes two core approximations: (1) layerwise independence approximation, where GNH is approximated as block-diagonal with each block corresponding to GNH of some specific layer, and (2) input activations $\mathbf{a}$ and pseudo-gradient of the pre-activations $\mathcal{D}\mathbf{s}$ are independent under the model's predictive distribution. The layerwise GNH can be approximated as:

$$\mathbf{G} = \mathbb{E}\left[ \text{vec}(\mathcal{D}\mathbf{W})\text{vec}(\mathcal{D}\mathbf{W})^\top \right] = \mathbb{E}\left[ \mathbf{a}\mathbf{a}^\top \otimes \mathcal{D}\mathbf{s}\mathcal{D}\mathbf{s}^\top \right] \approx \mathbb{E}[\mathbf{a}\mathbf{a}^\top] \otimes \mathbb{E}\left[ \mathcal{D}\mathbf{s}\mathcal{D}\mathbf{s}^\top \right] := \mathbf{A} \otimes \mathbf{S}, \tag{34}$$

where $\otimes$ denotes the Kronecker product. The matrices $\mathbf{A} \in \mathbb{R}^{I \times I}$ and $\mathbf{S} \in \mathbb{R}^{O \times O}$ in Equation (34) represent the uncentered covariance matrices of the activations and the pseudo-gradients with respect to the pre-activations, respectively. These covariance matrices can be estimated by computing the statistics over many data batches and taking the average.

Denoting the eigendecomposition of these covariance matrices as $\mathbf{A} = \mathbf{Q_A}\boldsymbol{\Lambda_A}\mathbf{Q_A}^\top$ and $\mathbf{S} = \mathbf{Q_S}\boldsymbol{\Lambda_S}\mathbf{Q_S}^\top$, using properties of the Kronecker product, we can express the eigendecomposition of $\mathbf{A} \otimes \mathbf{B}$ as:

$$\mathbf{A} \otimes \mathbf{B} = (\mathbf{Q_A} \otimes \mathbf{Q_S})(\boldsymbol{\Lambda_A} \otimes \boldsymbol{\Lambda_S})(\mathbf{Q_A} \otimes \mathbf{Q_S})^\top. \tag{35}$$

EK-FAC introduces a more accurate approximation to the GNH by introducing a compact representation of the eigenvalues (instead of representing them as the Kronecker product $\boldsymbol{\Lambda_A} \otimes \boldsymbol{\Lambda_S}$). The layerwise GNH for EK-FAC is represented as follows:

$$\mathbf{G} \approx (\mathbf{Q_A} \otimes \mathbf{Q_S})\boldsymbol{\Lambda}(\mathbf{Q_A} \otimes \mathbf{Q_S})^\top. \tag{36}$$

Here, the corrected eigenvalues $\boldsymbol{\Lambda} \in \mathbb{R}^{IO \times IO}$ are defined as:

$$\boldsymbol{\Lambda}_{ii} := \mathbb{E}[((\mathbf{Q_A} \otimes \mathbf{Q_S})\text{vec}(\mathcal{D}\mathbf{W}))_i^2]. \tag{37}$$

The corrected eigenvalues in Equation (37) minimize the approximation error with the GNH measured by the Frobenius norm, where we refer readers to George et al. [18] for the derivations.

## D.2 EK-FAC Computations for SOURCE

As detailed in Section 3.3, our practical instantiation of SOURCE requires averaging the Hessians across checkpoints within a segment. We use a common averaging scheme in the optimization literature [63, 18, 26] to compute the averaged EK-FAC factors. We first compute the activation covariance matrices $\mathbf{A}$ and pseudo-gradient covariance matrices $\mathbf{S}$ for all model checkpoints. These matrices are obtained by computing the statistics over all data points once (1 epoch). Then, we take the average over these covariance matrices to obtain $\bar{\mathbf{A}} = \frac{1}{C_\ell} \sum_{k=1}^{C_\ell} \mathbf{A}_k$ and $\bar{\mathbf{S}} = \frac{1}{C_\ell} \sum_{k=1}^{C_\ell} \mathbf{S}_k$, where $C_\ell$ is the total number of model checkpoints for the $\ell$-th segment and $\mathbf{A}_k$ and $\mathbf{S}_k$ are covariance matrices for the $k$-th checkpoint. Then, we perform eigendecomposition on these averaged covariance matrices to obtain the eigenvectors $\bar{\mathbf{Q}}_\mathbf{A}$ and $\bar{\mathbf{Q}}_\mathbf{S}$. Under the eigenbasis $\bar{\mathbf{Q}}_\mathbf{A} \otimes \bar{\mathbf{Q}}_\mathbf{S}$, we compute the corrected eigenvalues $\mathbf{\Lambda}_k$ for each model checkpoint (Equation (37)) and then average the eigenvalues to obtain $\bar{\mathbf{\Lambda}}$. In summary, the averaged (Gauss-Newton) Hessian for a particular segment is approximated as:

$$\bar{\mathbf{G}} \approx (\bar{\mathbf{Q}}_\mathbf{A} \otimes \bar{\mathbf{Q}}_\mathbf{S}) \bar{\mathbf{\Lambda}} (\bar{\mathbf{Q}}_\mathbf{A} \otimes \bar{\mathbf{Q}}_\mathbf{S})^\top. \tag{38}$$

SOURCE requires computing the covariance matrices and corrected eigenvalues for each model checkpoint. Moreover, calculating the TDA scores for all training data points requires computing the training gradients $C$ times, where $C$ is the total number of checkpoints. Hence, SOURCE is approximately $C$ times more computationally expensive than influence functions evaluated at the final checkpoint. In Section 3.3, we introduced a more efficient variant, which averages the parameters within a segment instead. This variant only needs to compute the EK-FAC factors once for each segment and requires computing the EK-FAC factors and gradients $L$ times. Hence, it is $L$ times more computationally expensive than influence functions.

When the model is trained with SGD with a heavy ball momentum $\beta$ (SGDm), we scaled the learning rate used in SOURCE as $\bar{\eta}_\ell (1 - \beta)^{-1}$ to account for the effective learning rate (terminal velocity). In cases where AdamW optimizers are used as in Appendix C, computing the matrix exponential for $\bar{\mathbf{P}}_\ell^{1/2} \bar{\mathbf{H}}_\ell \bar{\mathbf{P}}_\ell^{1/2}$ is challenging with EK-FAC. We additionally keep track of the diagonal Hessian approximation (which can be easily and efficiently obtained when computing the corrected eigenvalues in Equation (37)) and use the diagonal Hessian approximation for computing the matrix exponential in Equation (23) and Equation (27). Note that we still use the EK-FAC factors to compute $\bar{\mathbf{H}}_\ell^{-1} \bar{\mathbf{g}}_\ell$ in Equation (27).

## D.3 Applicability to Other Approximation Techniques

While we described one instantiation of SOURCE with the EK-FAC approximation, SOURCE can be integrated with other techniques used for approximating implicit-differentiation-based TDA methods, such as TRAK [70] and DATAINF [55]. For example, as in TRAK, we can use random projection [43] to efficiently compute the averaged Hessian and gradients in a lower-dimensional space. TRAK is advantageous over the EK-FAC approximation when there are many query data points, as it caches compressed training gradients in memory, avoiding recomputing them for each query.

# E Experimental Setup

This section describes the experimental setup used to obtain the results presented in Section 5. This includes a description of each task (Appendix E.1) and the methodology for computing the linear datamodeling score (LDS) (Appendix E.2). Implementation details of the subset removal counterfactual evaluation and baseline techniques are provided in Appendix E.3 and Appendix E.4, respectively. All experiments were conducted using PYTORCH version 2.1.0 [71]. We used CPUs to conduct UCI regression experiments, A100 (80GB) GPUs to conduct GLUE and WikiText-2 experiments, and A6000 (48GB) GPUs for other experiments. A single GPU was used to run SOURCE and baseline techniques. The internal cluster was used to run the experiments. Since our ground truth requires a lot of model retraining with different random choices (*e.g.*, initialization and batch ordering), we considered tasks that train (or fine-tune) with less than 20 minutes using the abovementioned compute resources. For example, the generation of the LDS ground truth for a given task and data sampling ratio $\alpha \in (0, 1)$ takes at most 210 hours of computational resources.

## E.1 Datasets and Models

We conducted systematic hyperparameter optimization for all tasks. This process involved conducting grid searches to find hyperparameter configurations that achieve the best average validation performance (accuracy for classification tasks and loss for others). The average validation performance was obtained by retraining the network 5 times using different random seeds. For models trained with SGD with a heavy ball momentum of 0.9 (SGDm), our search spaces for learning rate and weight decay were {3e-1, 1e-1, 3e-2, 1e-2, 3e-3, 1e-3, 3e-4, 1e-4, 3e-5, 1e-5} and {3e-2, 1e-2, 3e-3, 1e-3, 3e-4, 1e-4, 3e-5, 1e-5, 0.0}, respectively. For models trained with AdamW [61], the search spaces were {1e-2, 3e-3, 1e-3, 3e-4, 1e-4, 3e-5, 1e-5} for learning rate and {3e-2, 1e-2, 3e-3, 1e-3, 3e-4, 1e-4, 3e-5, 1e-5, 0.0} for weight decay. In cases where the original experimental setup from which we adapted had a pre-specified learning rate and weight decay, these hyperparameters were incorporated into our search space.

**UCI Datasets (Regression).**   For regression tasks, we used the Concrete [94] and Parkinson [83] datasets from the UCI Machine Learning Repository [45]. Both datasets were pre-processed to have a zero mean and unit variance for input features and targets. We trained a three-layer multilayer perceptron (MLP), where each layer consisted of 128 hidden units and the RELU activation function. The models were optimized using SGDm for 20 epochs with a batch size of 32 and a constant learning rate schedule. A learning rate of 3e-2 and a weight decay of 1e-5 were used for the Concrete dataset. For the Parkinson dataset, the learning rate was set to 1e-2 with a weight decay value of 3e-5. We saved 6 intermediate checkpoints throughout training. For the noisy Concrete (Concrete-N) dataset, we randomly modified 30% of the targets by sampling from a Normal distribution with zero mean and unit variance. We used the same hyperparameters but trained the models for 3 epochs.

**MNIST & FashionMNIST (Image Classification).**   Following the experimental setup from Koh and Liang [49] and Bae et al. [3], we trained a three-layer multilayer perceptron (MLP) on approximately 10% of MNIST [57] and FashionMNIST [91] datasets. Smaller versions of these datasets were used to compute the counterfactual ground truth more efficiently. The models were trained with SGDm for 20 epochs with a batch size of 64 and a constant learning rate. The learning rate and weight decay were set for both datasets to 3e-2 and 1e-3, respectively. We saved 6 checkpoints during training and utilized them for TRACIN and SOURCE. For the noisy FashionMNIST (FashionMNST-N) experiment, we randomly relabeled 30% of the training dataset. The network was only trained for 3 epochs with a learning rate 1e-2 and weight decay 3e-5.

**CIFAR-10 (Image Classification).**   For the CIFAR-10 dataset [53], we trained the ResNet-9 model [32],[8] following the standard data augmentation procedure from Zagoruyko and Komodakis [95]. This included extracting images from a random $32 \times 32$ crop after applying zero-padding of 4 pixels, with a 50% probability of horizontal flipping. The network was trained for 25 epochs using SGDm with a batch size of 512 and a cyclic learning rate schedule, peaking at 0.5. The initial learning rate was set to 0.4 with a weight decay of 1e-3, and 6 intermediate checkpoints were saved throughout training.

**GLUE (Text Classification).**   We fine-tuned the BERT model [12] on SST-2, RTE, and QNLI datasets from the GLUE benchmark [85] with the training script from the `Transformers` library [89].[9] Following the experimental setup from Park et al. [70], we capped the training dataset at a maximum of 51200 examples to compute the LDS efficiently. However, we did not modify the original architecture (*e.g.*, removing the last TANH layer) and trained the network with the AdamW optimizer. The weight decay was set to 1e-2 for all tasks, and the learning rates were set as follows: 3e-5 for SST-2, 1e-5 for QNLI, and 2e-5 for RTE. We saved 6 intermediate checkpoints for each training run.

**WikiText-2 (Language Modeling).**   For the language modeling task, we fine-tuned the GPT-2 model [73] using the WikiText-2 dataset [65]. We followed the training script from the Transformer

---

[8] `https://github.com/MadryLab/trak/blob/main/examples/cifar_quickstart.ipynb`.
[9] `https://github.com/huggingface/transformers/blob/main/examples/pytorch/text-classification/run_glue_no_trainer.py`.

library but set the maximum sequence length to $512$.[10] During fine-tuning with AdamW, we saved 6 intermediate checkpoints for data attribution. The learning rate, weight decay, and batch size were set to 3e-5, 1e-2, and 8, respectively. We set the measurement $f$ as a loss for the language modeling task.

**RotatedMNIST & PACS (Image Classification).** We used the RotatedMNIST dataset [20] and the PACS dataset [58], following the data pre-processing procedures from Gulrajani and Lopez-Paz [25].[11] The training process was divided into two distinct stages for both tasks. During the initial stage of the training, we trained the network with the dataset $\mathcal{D}_1$, while the second stage used dataset $\mathcal{D}_2$. For RotatedMNIST, the first dataset $\mathcal{D}_1$ was comprised of images rotated at 0, 15, 45, and 60 degrees, whereas the second dataset $\mathcal{D}_2$ contained images rotated at 30 degrees. We trained a three-layer MLP for 30 (20/10) epochs using SGDm and a batch size of 128. The learning rate and weight decay were set to 1e-1 and 1e-5. For PACS, the first dataset $\mathcal{D}_1$ included images from the cartoon, photo, and sketch categories, and the second dataset $\mathcal{D}_2$ had art paintings. We fine-tuned ResNet-50 [32], initialized from the pre-trained parameters,[12] using SGDm for 40 (30/10) epochs with a batch size of 128, a learning rate of 1e-4, and a weight decay of 3e-5.

## E.2 Linear Datamodeling Score

We follow a methodology proposed by Park et al. [70] to compute the linear datamodeling score (LDS). Let $\boldsymbol{\lambda}$ represent the set of hyperparameters used for training the model on a specified task, such as the choice of optimizer and the number of training epochs. Let $\alpha \in (0, 1)$ denote the data sampling ratio. The process for obtaining the LDS involves several steps:

1. We generate $M$ data subsets, denoted as $\{\mathcal{S}_j\}_{j=1}^M$, each being a uniformly sampled subset of the original training dataset $\mathcal{D}$. Each subset $\mathcal{S}_j \subset \mathcal{D}$ contains $\lceil \alpha N \rceil$ data points, where $N$ denotes the total number of training data points.

2. For each data subset $\mathcal{S}_j$, the model is trained $R$ times using different random seeds $\{\xi_r\}_{r=1}^R$ (*e.g.*, model initialization and batch ordering).

3. Given an attribution method $\tau$ and a query example $\boldsymbol{z}_q$, we measure the Spearman correlations [81] between the prediction and the estimated expected measurable quantity:

$$\boldsymbol{\rho}\left(\left\{\frac{1}{R}\sum_{r=1}^R f(\boldsymbol{z}_q, \boldsymbol{\theta}^s(\mathcal{S}_j; \boldsymbol{\lambda}, \xi_r)) : j \in [M]\right\}, \{g_\tau(\boldsymbol{z}_q, \mathcal{S}_j, \mathcal{D}; \boldsymbol{\lambda}) : j \in [M]\}\right), \quad (39)$$

where $g$ represents the group attribution prediction, expressed as:

$$g_\tau(\boldsymbol{z}_q, \mathcal{S}, \mathcal{D}; \boldsymbol{\lambda}) := \sum_{\boldsymbol{z} \in \mathcal{S}} \tau(\boldsymbol{z}_q, \boldsymbol{z}, \mathcal{D}; \boldsymbol{\lambda}). \quad (40)$$

4. To obtain the final LDS, we average the correlations over a set of query data points (up to 2000 in our experiments) and report the score with 95% bootstrap confidence intervals, which accounts for resampling of the data subset $\mathcal{S}_j$ (see Park et al. [70] for details).

For a given data sampling ratio $\alpha$, the networks must be retrained $MR$ times in total to compute the LDS ground truth. In our experiments, we used 100 subsets ($M = 100$). The repeat $R$ was set to 100 for UCI regression tasks, 10 for MNIST classification tasks, 20 for CIFAR-10 image classification task, 5 for GLUE text classification and WikiText language modeling task, and 20 for RotatedMNIST and PACS image classification tasks. We used the largest feasible $R$ based on our computational budget because we observed improvements in LDS for baseline techniques (especially TRAK, IF, and SOURCE) with larger $R$.

## E.3 Subset Removal Counterfactual Evaluation

For the subset removal counterfactual evaluation, we first train the model with the full dataset $\mathcal{D}$ under different random choices (over 5 random seeds) and select 100 test data points correctly classified on

---

[10]`https://github.com/huggingface/transformers/blob/main/examples/pytorch/language-modeling/run_clm_no_trainer.py`.

[11]`https://github.com/facebookresearch/DomainBed`.

[12]`https://pytorch.org/vision/main/models/generated/torchvision.models.resnet50.html`.

all random choices. Then, for each test data point and attribution technique, we remove the top-$k$ data points from the pre-defined interval $k_1, \ldots, k_I$ (such that $k_1 < \cdots < k_I$), as indicated as highly positively influential by the data attribution technique, retrain the network with this modified dataset, and examine if the original test data point gets misclassified on average under different random choices (over 3 random seeds). Finally, for each value of $k$ in the pre-defined interval, we report the fraction of test data points that get misclassified after removing at most top-$k$ training data points and retraining the network with the modified dataset.

For each TDA technique, this process requires retraining the model $100 \times I \times 3$ times, where $I$ is the pre-defined interval size. We set $I = 6$ for all experiments, leading to the retraining of the model 1800 times. To reduce the computational cost, we start from the smallest subset removal size $k_1$, and if the test data point gets misclassified under the current subset, we do not consider it for the larger subset removal size (*e.g.*, $k_2$). Hence, this can be seen as the fraction of test data points that get misclassified by removing at most $k$ training data points (evaluated at a fixed interval). We note that Singla et al. [79] instead use a bisection search to find the smallest subset size in which a test data point can be misclassified, whereas Ilyas et al. [35] use more fine-grained intervals with more number of seeds (*e.g.*, 8 intervals and 20 seeds). While it is possible to use a larger number of seeds (20 seeds as in Ilyas et al. [35]), because of computational limitations, we use 3 seeds to estimate the averaged misclassification. We used SOURCE and baseline techniques described in Appendix E.4 to identify positively influential training data points. We also included a RANDOM baseline, where we removed the training data points belonging to the same class as the target test example.

## E.4  Baselines

This section describes the baseline techniques used in Section 5. Unless specified otherwise, we describe them in the context of a single-training-run estimator, where the TDA techniques use the final parameters $\boldsymbol{\theta}^s$ obtained with hyperparameters $\boldsymbol{\lambda}$ and some random choice $\xi$ (the multiple-training-runs estimators simply average the TDA scores obtained from models trained with different random choices $\xi$).

**Representation Similarity (REPSIM).**  Representation similarity technique [9] evaluates the importance of a training data point $\boldsymbol{z}_m \in \mathcal{D}$ to a specific query data point $\boldsymbol{z}_q$ by comparing the latent representations of these data point pairs. This can be formulated as follows:

$$\tau_{\text{REPSIM}}(\boldsymbol{z}_q, \boldsymbol{z}_m, \mathcal{D}; \boldsymbol{\lambda}) \coloneqq \text{similarity}(\phi_{\boldsymbol{\theta}^s}(\boldsymbol{z}_q), \phi_{\boldsymbol{\theta}^s}(\boldsymbol{z}_m)). \tag{41}$$

Here, similarity$(\mathbf{v}_1, \mathbf{v}_2)$, where $\mathbf{v}_1$ and $\mathbf{v}_2$ are some vectors, is typically defined through the $\ell_2$ metric, dot metric, or cosine metric [30]. In our experiments, the function $\phi_{\boldsymbol{\theta}^s}(\boldsymbol{z})$ was designed to map a data point to its last hidden activations (before the final output layer), using a forward pass through the final parameters $\boldsymbol{\theta}^s$. We used the cosine metric to compute the attribution score but observed similar performance when using the $\ell_2$ metric, aligning with observations in previous studies [35, 70, 79].

**TRACIN.**  We used the TRACINCP estimator from Pruthi et al. [72], defined as:

$$\tau_{\text{TRACIN}}(\boldsymbol{z}_q, \boldsymbol{z}_m, \mathcal{D}; \boldsymbol{\lambda}) \coloneqq \sum_{k=1}^{C} \eta_k \cdot \nabla_{\boldsymbol{\theta}} f(\boldsymbol{z}_q, \hat{\boldsymbol{\theta}}_k) \cdot \nabla_{\boldsymbol{\theta}} \mathcal{L}(\boldsymbol{z}_m, \hat{\boldsymbol{\theta}}_k), \tag{42}$$

where $C$ represents the number of checkpoints, $\hat{\boldsymbol{\theta}}_k$ represents the parameters at the $k$-th checkpoint, and $\eta_k$ is the learning rate applied at the corresponding checkpoint. The last checkpoint is typically set to the final model parameters $\boldsymbol{\theta}^s$. While there is an option to compress the gradients using a random projection as suggested by Pruthi et al. [72], our experiments used the full gradients to obtain a stronger baseline. The checkpoint selection details are described in Appendix E.1.

**Influence Functions (IF).**  As detailed in Section 2.2, training data attribution with influence functions is formulated as follows:

$$\tau_{\text{IF}}(\boldsymbol{z}_q, \boldsymbol{z}_m, \mathcal{D}; \boldsymbol{\lambda}) \coloneqq \nabla_{\boldsymbol{\theta}} f(\boldsymbol{z}_q, \boldsymbol{\theta}^s)^\top \mathbf{H}^{-1} \nabla_{\boldsymbol{\theta}} \mathcal{L}(\boldsymbol{z}_m, \boldsymbol{\theta}^s), \tag{43}$$

where $\mathbf{H}$ denotes the Hessian of the cost at the final parameters $\boldsymbol{\theta}^s$. To make influence functions scalable to large neural networks, we used the Eigenvalue-corrected Kronecker-Factored Approximate Curvature (EK-FAC) parameterization [18] to approximate the Hessian, as proposed by Grosse et al.

[24]. We refer readers to Grosse et al. [24] and Appendix D for details on the EK-FAC computation. Relatedly, Schioppa et al. [76] use Arnoldi iterations [2], and Kwon et al. [55] utilize the parameter-efficient fine-tuning (PEFT) [34] strategy to efficiently approximate influence functions. More recently, Choe et al. [11] further utilized low-dimensional gradient projection to compute influence functions more efficiently.

While Grosse et al. [24] only consider the computation of influence scores to the MLP layers of transformers [84], in our experiments, we extended this computation to include the attention layers as well. We excluded layer normalization, batch normalization, and embedding layers from the influence computation. Influence functions have an additional hyperparameter $\lambda > 0$, which is used to compute the damped inverse Hessian-vector product (IHVP), denoted as $(\mathbf{H} + \lambda\mathbf{I})^{-1}\mathbf{v}$ for some vector $\mathbf{v}$. We used a small damping term for consistency with TRAK [70] and set it to 1e-8 to avoid numerical instability (note that TRAK sets the damping term to 0).

**TRAK.**   In contrast to the traditional formulation of influence functions, TRAK [70] leverages random projections [43], Generalized Gauss-Newton approximation, and ensembling for data attribution. Specifically, given a random projection matrix $\mathbf{P} \sim \mathcal{N}(0, 1)^{M \times K}$, where $K$ denotes the projection dimension, the final model parameters $\boldsymbol{\theta}^s$, and a model output function $f(\boldsymbol{z}, \boldsymbol{\theta})$, TRAK projects all training and query gradients into $K$-dimensional vectors. The feature map is defined as:

$$\phi(\boldsymbol{z}) \coloneqq \mathbf{P}^\top \nabla_{\boldsymbol{\theta}} f(\boldsymbol{z}, \boldsymbol{\theta}^s). \tag{44}$$

We further define $\boldsymbol{\Phi} \coloneqq [\phi_1; \ldots; \phi_N] \in \mathbb{R}^{N \times K}$ as stacked projected gradients for all training data points, where each $\phi_i$ corresponds to $\phi(\boldsymbol{z}_i)$. Subsequently, TRAK's single model estimator is formulated as:

$$\tau_{\text{TRAK}}(\boldsymbol{z}_q, \cdot, \mathcal{D}; \boldsymbol{\lambda}) \coloneqq \phi(\boldsymbol{z}_q)^\top (\boldsymbol{\Phi}^\top \boldsymbol{\Phi})^{-1} \boldsymbol{\Phi}^\top \mathbf{Q}, \tag{45}$$

with $\mathbf{Q}$ being a $N \times N$ diagonal matrix for weightings. Here, $\tau_{\text{TRAK}}$ represents a vector of dimension $N$, containing attribution score for each training data point. TRAK uses an ensemble of single model estimators, each derived from models trained with distinct configurations and projection matrices. We refer readers to Park et al. [70] and Engstrom et al. [13] for detailed derivations and discussions of TRAK.

We used the final checkpoints for TRAK in our experimental setup involving a single model. We computed TRAK using the last checkpoint of 10 differently trained models (each trained with 50% of the dataset) for experiments with multiple model setups. TRAK has a hyperparameter that determines the dimension of the random projection $K$. We set the projection dimension to 20480 for ResNet-9 and RotatedMNIST, 8192 for ResNet-50 on the PACS dataset, 1024 for BERT trained on the RTE dataset and 512 for MLP trained on the Concrete dataset (due to the datasets' smaller size), and 4096 for all other tasks. All experiments were conducted using TRAK's official implementation.[13]

**Empirical Influence (EI).**   To compute the empirical influence (DOWNSAMPLING) [17], we first create $M$ data subsets $\{\mathcal{S}_j\}_{j=1}^M$, each being a uniformly sampled subset of the original training dataset. Each subset $\mathcal{S}_i$ contains $\lceil \alpha N \rceil$ data points, where $\alpha \in (0, 1)$ is the data sampling ratio. Given a training data point $\boldsymbol{z}_m \in \mathcal{D}$, we define $M_m$ as the total number of data subsets containing $\boldsymbol{z}_m$. The empirical influence scores are formulated as follows:

$$\tau_{\text{EI}}(\boldsymbol{z}_q, \boldsymbol{z}_m, \mathcal{D}; \boldsymbol{\lambda}) \coloneqq \frac{1}{M - M_m} \sum_{j=1}^M \mathbb{1}[\boldsymbol{z}_m \notin \mathcal{S}_j] f(\boldsymbol{z}_q, \boldsymbol{\theta}^s(\mathcal{S}_j; \boldsymbol{\lambda}, \xi_j)) \tag{46}$$

$$- \frac{1}{M_m} \sum_{j=1}^M \mathbb{1}[\boldsymbol{z}_m \in \mathcal{S}_j] f(\boldsymbol{z}_q, \boldsymbol{\theta}^s(\mathcal{S}_j; \boldsymbol{\lambda}, \xi_j)), \tag{47}$$

where $\mathbb{1}[\cdot]$ is an indicator function to determine if the training data point $\boldsymbol{z}_m$ is contained in the $j$-th data subset $\mathcal{S}_j$. Intuitively, Equation (46) computes the averaged query measurement when data point $\boldsymbol{z}_m$ is not used in training, whereas Equation (47) computes the averaged measurement when the data point is used in training. Following Zheng et al. [97], we created 512 data subsets ($M = 512$) with a sampling ratio $\alpha = 0.5$, which requires retraining the model 512 times with 50% of training data points removed.

---

[13]https://github.com/MadryLab/trak.

| Methods | LDS | |
| --- | --- | --- |
| | Single Model | Multiple Models |
| REPSIM [9] | $0.03 \pm 0.02$ | $0.04 \pm 0.02$ |
| TRACIN [72] | $0.20 \pm 0.02$ | $0.21 \pm 0.03$ |
| TRAK [70] | $0.08 \pm 0.01$ | $0.26 \pm 0.00$ |
| IF [49, 24] | $0.30 \pm 0.01$ | $0.45 \pm 0.01$ |
| DOWNSAMPLING [17] | - | $0.11 \pm 0.02$ |
| HYDRA [10] | $0.16 \pm 0.02$ | $0.17 \pm 0.02$ |
| SOURCE with averaged parameters (**ours**) | $0.42 \pm 0.01$ | $0.48 \pm 0.02$ |
| SOURCE (**ours**) | $\mathbf{0.46 \pm 0.01}$ | $\mathbf{0.53 \pm 0.01}$ |

Table 1: LDS at $\alpha = 0.5$ for SOURCE ($L = 3$) and baseline TDA techniques (including DOWNSAM-PLING and HYDRA) on the FashionMNIST dataset. We show the $95\%$ bootstrap confidence intervals.

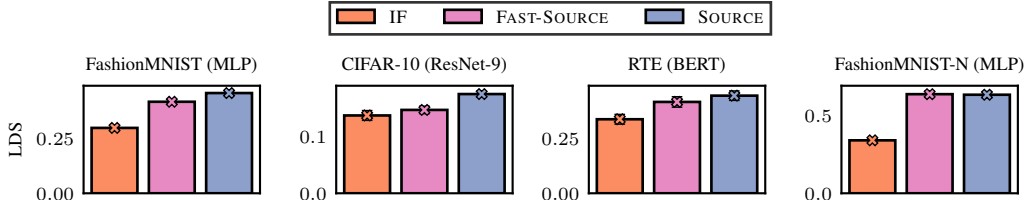

Figure 7: LDS at $\alpha = 0.5$ for influence functions, FAST-SOURCE (see Appendix F.2), and SOURCE. The LDS is shown for a single model (single-training-run) setup.

**HYDRA.** We used the fast version of HYDRA [10], formulated as:

$$\tau_{\text{HYDRA}}(\boldsymbol{z}_q, \boldsymbol{z}_m, \mathcal{D}; \boldsymbol{\lambda}) \coloneqq \sum_{k=0}^{T-1} \eta_k \cdot \mathbb{1}[\boldsymbol{z}_m \in \mathcal{B}_k] \cdot \nabla_{\boldsymbol{\theta}} f(\boldsymbol{z}_q, \boldsymbol{\theta}^s) \cdot \nabla_{\boldsymbol{\theta}} \mathcal{L}(\boldsymbol{z}_m, \boldsymbol{\theta}_k), \tag{48}$$

where $T$ represents the total number of gradient update steps, $\boldsymbol{\theta}_k$ denotes the parameters at the $k$-th iteration, and $\eta_k$ is the corresponding learning rate. Here, $\mathcal{B}_k$ denotes the batch of data points used at the corresponding update, and $\mathbb{1}[\boldsymbol{z}_m \in \mathcal{B}_k]$ is the indicator function for having selected $\boldsymbol{z}_m$ in the update. Note that HYDRA requires storing all parameter vectors used for training. We refer readers to Hammoudeh and Lowd [27] for derivations and detailed discussions of HYDRA.

# F  Additional Results

In this section, we present additional experimental results, including a comparison with additional baseline TDA techniques (Appendix F.1), an LDS evaluation of a computationally faster variant of SOURCE (Appendix F.2), an LDS evaluation at various sampling ratios and for more tasks (Appendix F.3), counterfactual evaluation on linear models (Appendix F.4), and visualizations of the top positively and negatively influential training data points for each TDA technique (Appendix F.5).

## F.1  Additional Baseline Comparisons

We compare SOURCE with empirical influence (DOWNSAMPLING) [17] and the fast version of HYDRA [10] on the FashionMNIST task. Results for these techniques on other tasks were omitted, since DOWNSAMPLING requires retraining the model over $500$ times and HYDRA necessitates saving all intermediate checkpoints throughout training. The implementation details are provided in Appendix E.4, and the results are shown in Table 1. SOURCE achieves the highest LDS on both single and multiple model setups compared to existing baseline TDA techniques we considered.

## F.2  SOURCE with Averaged Parameters

In Section 3.3, we introduced a more computationally efficient version of SOURCE, which averages the parameters within a segment instead of Hessians and gradients. Here, we present the LDS results at $\alpha = 0.5$ for the faster version, termed FAST-SOURCE, for FashionMNIST, CIFAR-10, RTE,

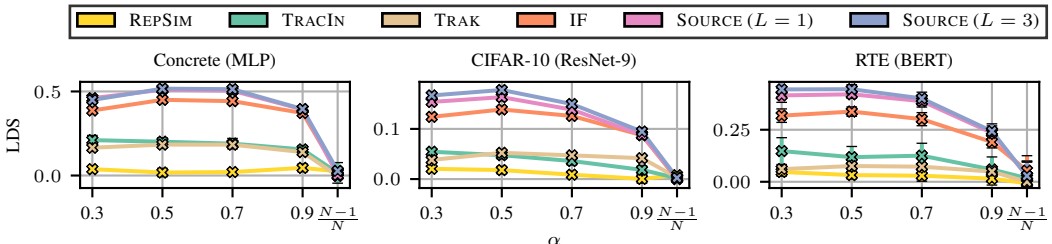

Figure 8: LDS across a range of data sampling ratios $\alpha$ for SOURCE ($L = \{1, 3\}$) and baseline TDA techniques. The LDS is measured for a single model setup, and error bars represent $95\%$ bootstrap confidence intervals.

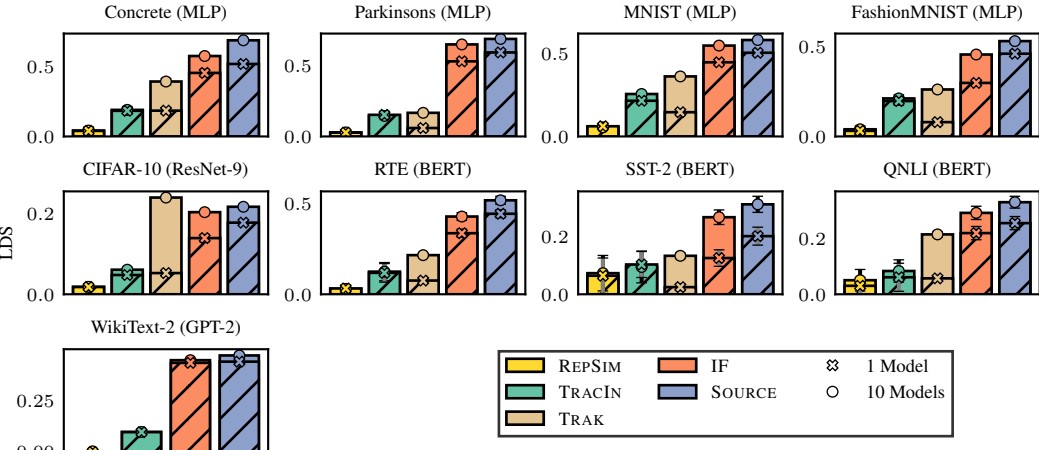

Figure 9: LDS at $\alpha = 0.5$ for SOURCE ($L = 3$) and baseline TDA techniques on models fully trained using a fixed dataset, where methods based on implicit differentiation and unrolling are expected to perform similarly. The error bars represent $95\%$ bootstrap confidence intervals. (Results for TRAK on WikiText-2 are omitted due to the lack of publicly available implementations for language modeling tasks.)

and FashionMNIST-N tasks. The results are shown in Figure 7. We observe that FAST-SOURCE outperforms influence functions on these tasks, while it generally achieves a lower LDS compared to SOURCE.

## F.3  Additional LDS Results

Here, we present additional results, evaluating TDA techniques with LDS at various sampling ratios and considering more tasks, where models are fully trained using a fixed dataset. We first consider computing the LDS across a range of data sampling ratios $\alpha$. (The procedures to compute the LDS are described in Appendix E.2.) The performance of SOURCE and other baseline attribution methods is shown in Figure 8. SOURCE consistently achieves higher LDS than the baseline methods across diverse $\alpha$ values. However, an exception is noted at $\alpha = 1 - 1/N$ (*e.g.*, removing a single training data point), where a significant drop in correlations is observed for all TDA methods. This finding is consistent with previous studies that highlight the limitations of LOO estimates in reliably evaluating attribution techniques [44, 14, 67] (see Appendix B for a detailed discussion). Additionally, our results suggest that while SOURCE with a single segment can be effective, using multiple segments typically improves LDS performance. Lastly, we observe that the relative rankings of TDA techniques typically remain consistent across various $\alpha$ values

Next, we present the LDS results at $\alpha = 0.5$ for additional tasks in Figure 9. SOURCE consistently outperforms baseline methods in a single model setup, achieving higher correlations with the ground truth. When aggregating TDA scores from multiple models, we observe a large improvement in the LDS. Our method obtains the highest LDS across all tasks, except for the CIFAR-10 classification task using ResNet-9. However, we show that our method outperforms baseline methods on the CIFAR-10 task for subset removal counterfactual evaluation in Section 5.2. We note that the tasks in Figure 9

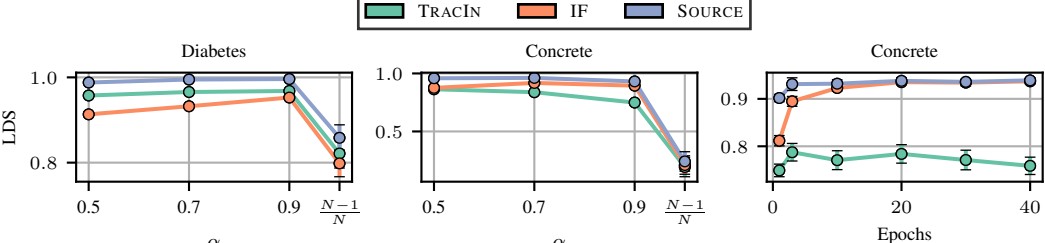

Figure 10: (Left & Middle) LDS for various values of data sampling ratios $\alpha$ on linear regression and logistic regression tasks trained for 3 epochs. (Right) LDS at $\alpha = 0.9$ for models trained with varying numbers of epochs. The error bars show $95\%$ bootstrap confidence intervals.

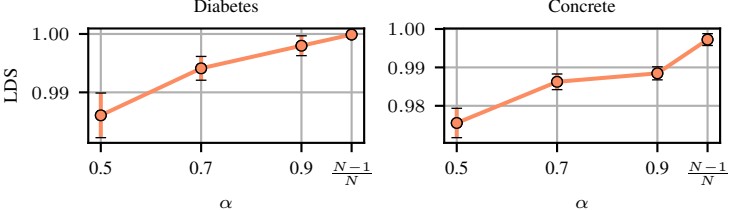

Figure 11: LDS on linear regression and logistic regression tasks for influence functions when TDA is performed on the optimal solution.

consider models sufficiently trained near convergence (with carefully chosen hyperparameters), where implicit-differentiation-based methods are expected to perform similarly to unrolling-based methods.

## F.4 Counterfactual Evaluations on Linear Models

In this section, we demonstrate the effectiveness of SOURCE on linear models when the model has not been trained until convergence. We trained linear regression on the Concrete dataset and logistic regression on the Diabetes dataset [80] for 3 epochs with a batch size of 32. We also constructed the LDS ground truth using SGD with the same hyperparameters. We applied TRACIN, IF, and SOURCE (with $L = 1$) to the trained model and computed the LDS for various data sampling ratios $\alpha$. The results are shown in Figure 10 (Left & Middle). SOURCE achieves higher LDS on all data sampling ratios for both regression and classification tasks. We further show the LDS at $\alpha = 0.9$ with varying numbers of epochs in Figure 10 (Right). (The LDS ground truth is recomputed at each epoch.) We observe a larger LDS gap between SOURCE and IF when the model was only trained for a small number of epochs, and the gap reduces as we train the model for a larger number of iterations. These results show that our formulation for SOURCE better supports TDA when the network has not fully converged, even in the case of linear models.

For completeness, we show the LDS for influence functions when the TDA is performed on the optimal solution in Figure 11. For each model, we computed the optimal solution (for logistic regression, we used the L-BFGS [59]), computed the influence function estimates, and evaluated their accuracy with LDS (also obtained by computing the optimal solution without some data points). As shown in Figure 11, influence functions obtain high correlations with the ground truth across various values of data sampling ratio $\alpha$. In contrast to neural network experiments in Appendix F.3, we observe an increase in the LDS as the data sampling ratio $\alpha$ increases (predicting the effect of removing a smaller number of data points), as the group influence predictions introduce more approximation error [3]. Notably, we obtain a high LDS when $\alpha = (N-1)/N$ (removing a single data point), as the LDS is computed at the precise optimal solution (see Appendix B for the discussion). TRACIN and SOURCE are not applicable in these contexts, as we computed the optimal solution with the direct solution or with L-BFGS, instead of with gradient descent.

## F.5 Qualitative Results

We first present the top positively and negatively influential data points obtained by each TDA technique on multiple model settings. Note that for these multiple model settings, REPSIM, TRACIN, TRAK, IF, and SOURCE use an ensemble of 10 models trained with different random choices. The

results for FashionMNIST, CIFAR-10, and RotatedMNIST are shown in Figure 12, Figure 13, and Figure 14, respectively. We also show the top positively and negatively influential data points on the CIFAR-10 dataset for a single model setup in Figure 15. In Table 2, we present the top positively and negatively influential data points obtained by SOURCE on the RTE dataset.

# G   Broader Impact & Limitations of SOURCE

**Broader Impact.**   Our paper focuses on improving training data attribution, especially in cases where traditional implicit-differentiation-based methods, such as influence functions, struggle. While our work does not have a direct societal impact, as it focuses on algorithmic improvements, there may be societal implications for improving TDA techniques. On the positive side, as shown in Grosse et al. [24], TDA techniques can be used to understand and debug the misbehavior of neural networks (*e.g.*, LLMs), which can promote trust in deploying machine learning systems. By identifying the training data points responsible for specific model behaviors, TDA can help improve model interpretability, fairness, and robustness. This, in turn, can lead to more reliable and equitable AI systems that benefit society. However, TDA techniques also allow for the analysis of the impact of training data points on trained models, which can be used for crafting data poisoning attacks [16, 38, 69]. Malicious actors could potentially use TDA to identify and manipulate influential data points, leading to the creation of biased or misleading models. This could have negative consequences, such as the spread of disinformation or unfair treatment of specific groups. To mitigate these risks, it is essential to develop responsible practices for using TDA techniques.

**Limitations.**   Compared to the influence function employing the same EK-FAC parameterization [24], the practical implementation of the SOURCE requires the computation of EK-FAC factors and gradients for all checkpoints (when performing TDA on all segments). Note that, when TDA is performed on one specific segment $\ell$, the gradients only need to be computed for checkpoints within a segment (instead of all checkpoints). Denoting the total number of checkpoints as $C$ and the total number of segments as $L$, SOURCE on all segments exhibits an approximate computational cost of $C$ times higher. Our experiments used configurations with $C \in \{3, 6\}$ and $L \in \{2, 3\}$. We also introduced a faster version of SOURCE in Appendix F.2, which directly averages the parameters instead of averaging the EK-FAC factors and gradients; the faster version is $L$ times computationally expensive compared to the EK-FAC influence functions.

Compared to implicit-differentiation-based TDA techniques, SOURCE requires access to intermediate checkpoints throughout the training process and corresponding hyperparameters such as learning rate, number of iterations, and preconditioning matrix. In cases where the details of the training process are not available, implicit-differentiation-based TDA techniques, such as TRAK [70] and influence functions [49, 24], may be preferable.

Moreover, SOURCE approximates the distributions of the Hessian and gradient as stationary within each segment of the training trajectory. In certain scenarios, this may not be a reasonable approximation. For instance, when pre-training large transformer models, the Hessian or gradients may undergo drastic changes throughout the training process. If the stationarity approximation is too inaccurate, one can enhance the fidelity of SOURCE by dividing the training trajectory into a larger number of segments, albeit at the cost of increased computational requirements. While we used a fixed number of segments and checkpoints, partitioned equally at the early, middle, and late stages of training, we can extend SOURCE by automatically determining when to segment by examining the changes in the Hessian or gradients, which we leave for future work. Lastly, SOURCE approximate the Hessians and gradients at different time steps as statistically independent to obtain a tractable approximation for the expected total derivative in Section 3.2. As discussed, this independence approximation amounts to neglecting the autocorrelation of optimization iterates.

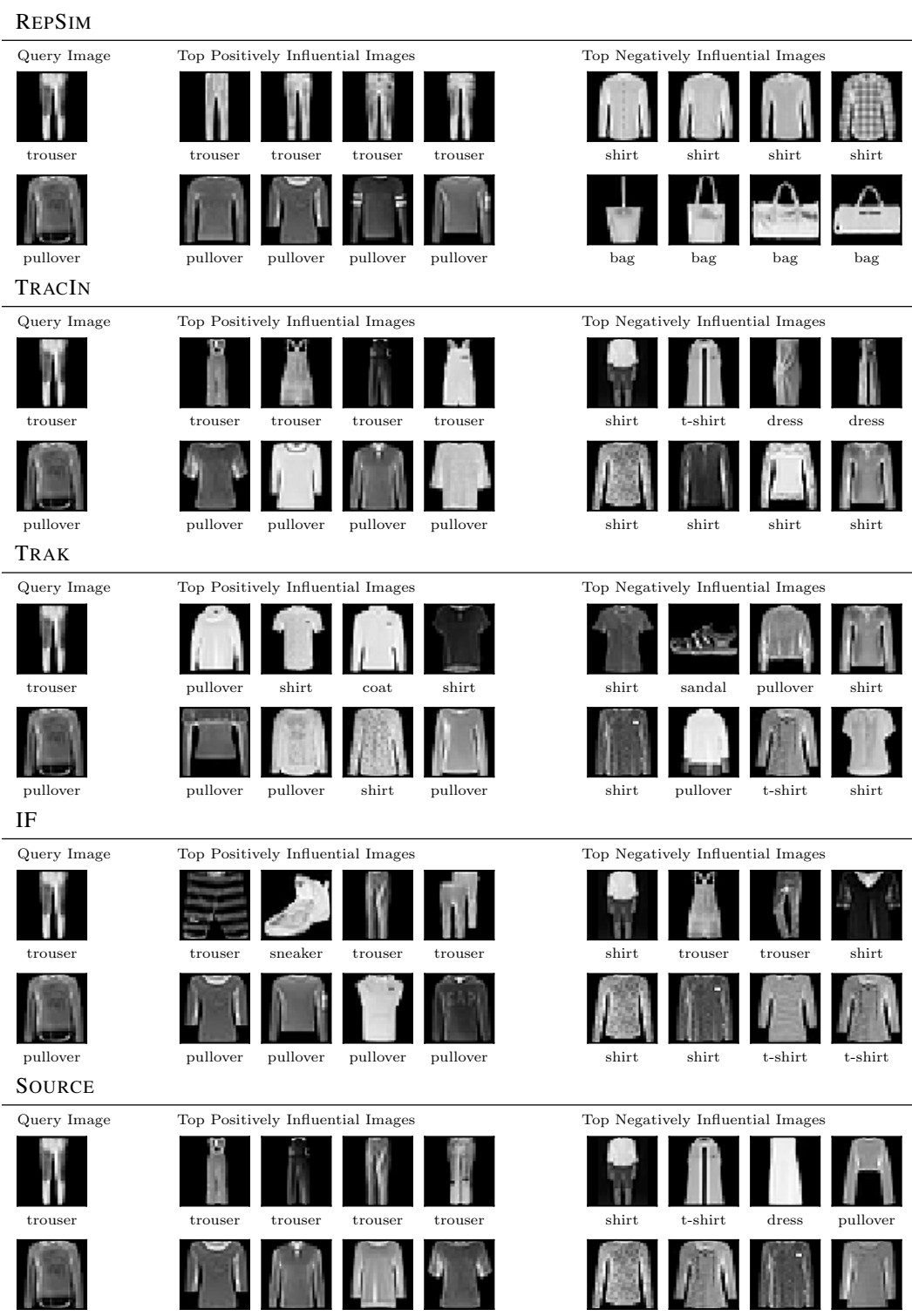

Figure 12: Top positively and negatively influential training images identified by SOURCE and baseline TDA techniques on the FashionMNIST dataset.

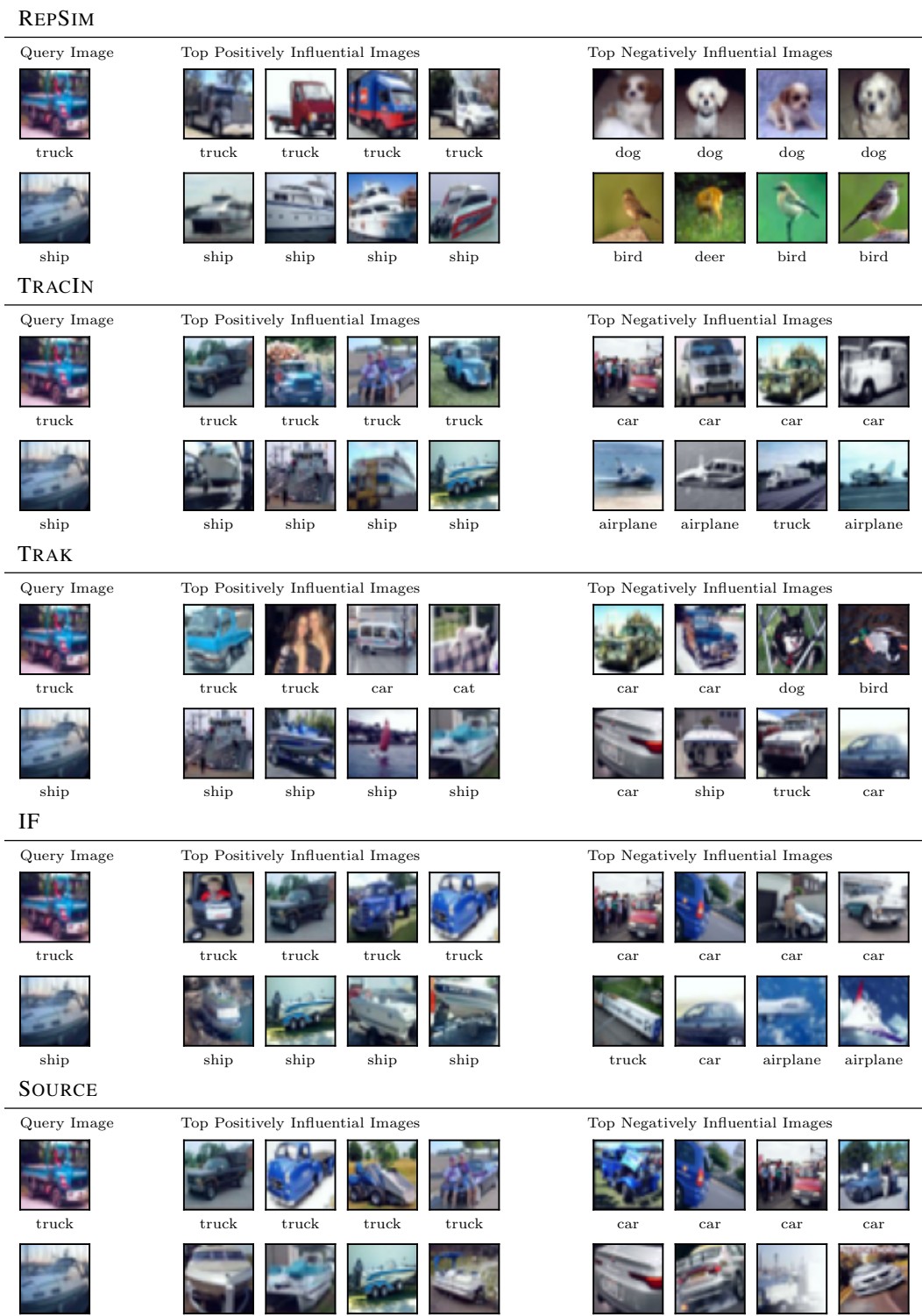

Figure 13: Top positively and negatively influential training images identified by SOURCE and baseline TDA techniques on the CIFAR-10 dataset. Note that we labeled the "automobile" class as "car".

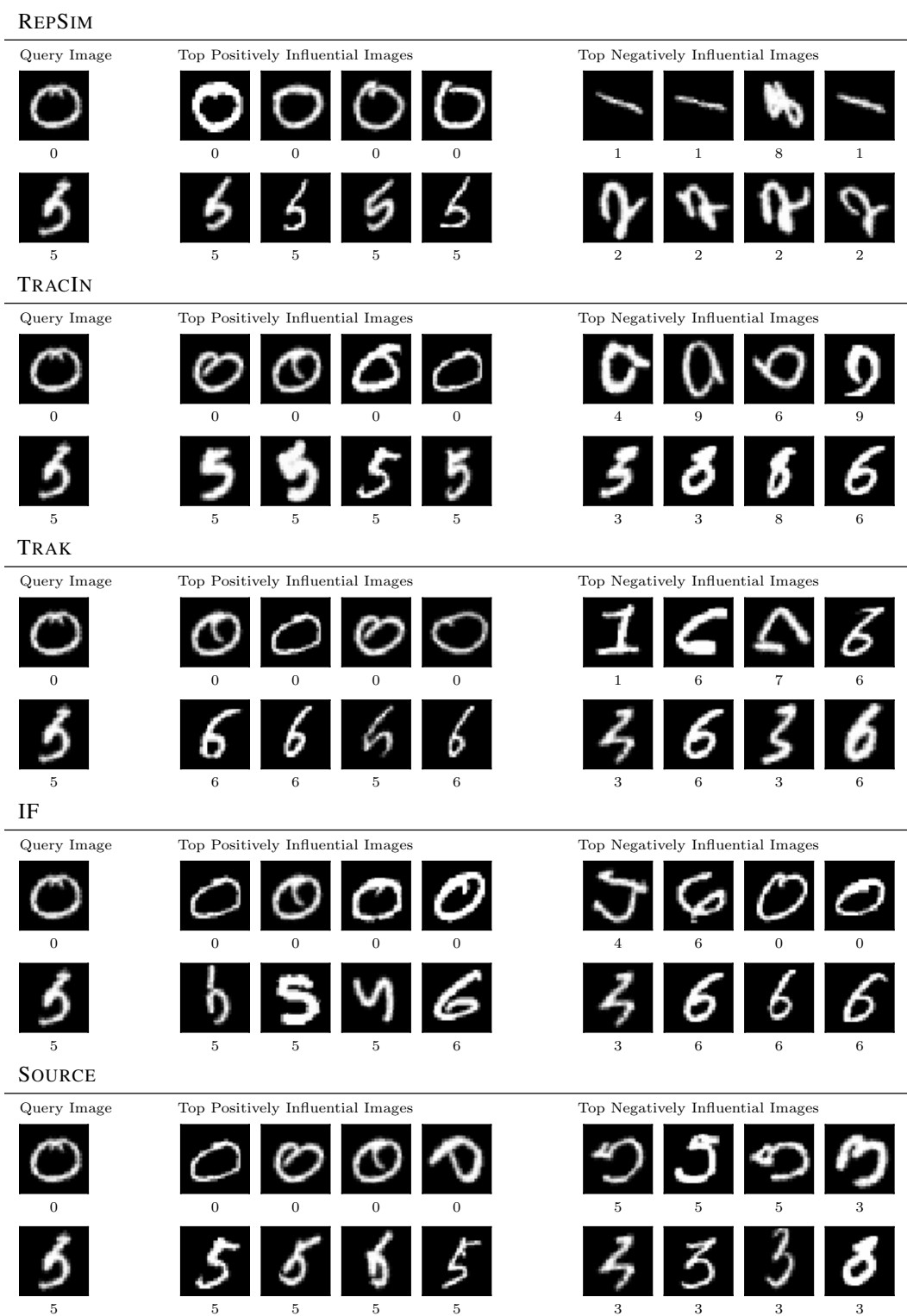

Figure 14: Top positively and negatively influential training images identified by SOURCE and baseline TDA techniques on the RotatedMNIST dataset.

| Query Data Point | Top Positively Influential Data Point | Top Negatively Influential Data Point |
|---|---|---|
| Dana Reeve, the widow of the actor Christopher Reeve, has died of lung cancer at age 44, according to the Christopher Reeve Foundation. / Christopher Reeve had an accident. **(not entailment)** | Though fearful of a forthcoming performance evaluation by her boss, Zoe must unravel the life of a man just found dead of a heart attack, who was supposed to have died three years earlier in a boating accident. / Zoe died in a boating accident. **(not entailment)** | Actor Christopher Reeve, best known for his role as Superman, is paralyzed and cannot breathe without the help of a respirator after breaking his neck in a riding accident in Culpeper, Va., on Saturday. / Christopher Reeve had an accident. **(entailment)** |
| Yet, we now are discovering that antibiotics are losing their effectiveness against illness. Disease-causing bacteria are mutating faster than we can come up with new antibiotics to fight the new variations. / Bacteria is winning the war against antibiotics. **(entailment)** | The papers presented show that all European countries are experiencing rapidly aging populations that will cause sharp increases in the cost of retirement income over the next several decades. / National pension systems currently adopted in Europe are in difficulties. **(entailment)** | Humans have won notable battles in the war against infection - and antibiotics are still powerful weapons - but nature has evolution on its side, and the war against bacterial diseases is by no means over. / Bacteria is winning the war against antibiotics. **(not entailment)** |
| Security forces were on high alert after an election campaign in which more than 1,000 people, including seven election candidates, have been killed. / Security forces were on high alert after a campaign marred by violence. **(entailment)** | Police sources stated that during the bomb attack involving the Shining Path, two people were injured. / Two people were wounded by a bomb. **(entailment)** | Pakistan President Pervez Musharraf has ordered security forces to take firm action against rioters following the assassination of opposition leader Benazir Bhutto. The violence has left at least 44 people dead and dozens injured. Mr. Musharraf insisted the measures were to protect people. VOA's Ayaz Gul reports from Islamabad that a bitter dispute has also erupted over how the 54-year-old politician died and who was behind her assassination. / Musharraf has ordered rioters to take firm action against security forces. **(not entailment)** |
| In 1979, the leaders signed the Egypt-Israel peace treaty on the White House lawn. Both President Begin and Sadat received the Nobel Peace Prize for their work. The two nations have enjoyed peaceful relations to this day. / The Israel-Egypt Peace Agreement was signed in 1979. **(entailment)** | Following the Israel-Egypt Peace Treaty of 1979, Israel agreed to withdraw from the Sinai Peninsula, in exchange for peace with its neighbor. For over two decades, the Sinai Peninsula was home to about 7,000 Israelis. / The Israel-Egypt Peace Agreement was signed in 1979. **(entailment)** | Canada and the United States signed an agreement on January 30, 1979, to amend the treaty to allow subsistence hunting of waterfowl. / The Israel-Egypt Peace Agreement was signed in 1979. **(not entailment)** |

Table 2: Top positively and negatively influential data points identified by SOURCE on the RTE dataset. A data point in the RTE dataset consists of a pair of sentences (separated by a forward slash "/") and a label indicating whether the second sentence entails the first sentence (entailment) or not (not entailment).

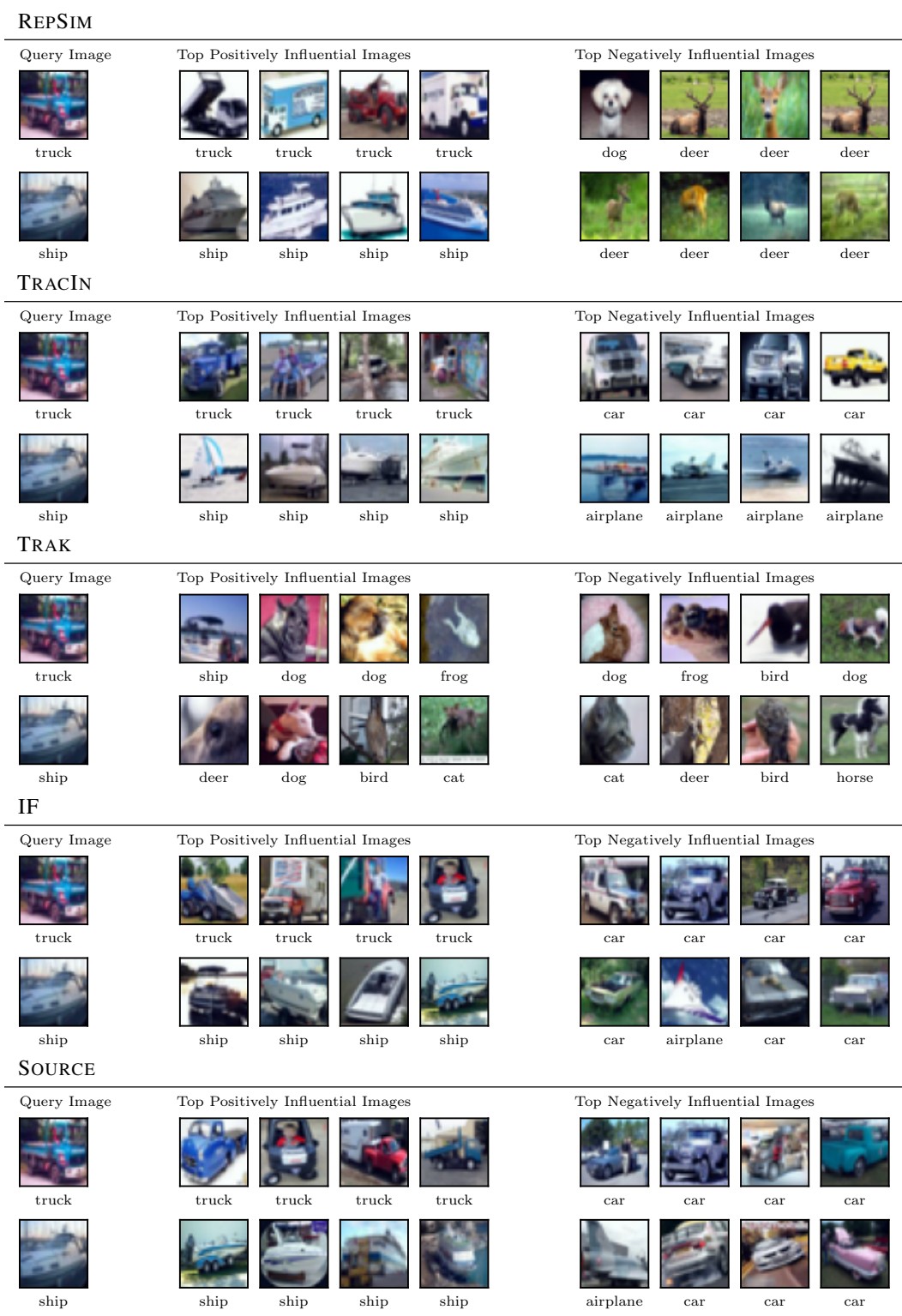

Figure 15: Top positively and negatively influential training images identified by SOURCE and baseline TDA techniques (single model setting) on the CIFAR-10 dataset.

