# OpenReview forum: "Training Data Attribution via Approximate Unrolling"
_NeurIPS.cc/2024/Conference — NeurIPS 2024 poster_

### Official Review · Reviewer_5ybi · 2024-07-09

**Soundness:** 4
**Presentation:** 4
**Contribution:** 4
**Rating:** 8
**Confidence:** 3

**Summary:**

The paper proposes a training data attribution (TDA) method based on implicit differentiation and unrolling. The goal is to estimate the effect of removing (or changing the weight) of a training example on the final (not necessarily optimal) parameters, accounting for training details influencing the trajectory of the parameters during training.

The method approximates the change in the final parameters $\boldsymbol\theta_T$ ($T$ is the number of optimization iterations) based on approximating the effect of re-weighting a training example with weight $1+\epsilon$. This requires computation of $\partial \boldsymbol\theta_T/\partial\epsilon$, which is a product of Jacobians $\partial\boldsymbol\theta_{k}/\partial\boldsymbol\theta_{k-1}$ from all training steps $k$., each computed from a Hessian of the loss. To reduce computational requirements, the authors apply different approximations while accounting for all parts of the training trajectory. The resulting algorithm, SOURCE, requires 2-18 more computation (Hessian estimations) than influence functions.

Experiments show that SOURCE outperforms other TDA methods in counterfactual evaluations on linear datamodeling score (LDS) (Park et al., 2023) and on standard subset removal counterfactual evaluation.

**Strengths:**

- The paper is very well written and structured.
- The paper is technically sound (as far as I have checked) and presents a valuable analysis and novel insights.
- The proposed training data attribution method, SOURCE, outperforms other TDA methods on counterfactual evaluations on different tasks across different data modalities.
- The SOURCE code will be published (if I interpret L692 correctly).

**Weaknesses:**

There are a few easy-to-fix errors:
- L6: missing comma before but.
- $\mathcal B_k$ is not mentioned before L124.
- L206-207: "approaches to".
- Appendix G is not referenced in the main text.
- L1046: "the SOURCE".
- L1051: $C = \{3, 6\}$ -> $C \in \{3, 6\}$ (likewise for $L$).
- (The curly brackets in Fig. 3 have a slight distortion.)

**Questions:**

**Questions:**
1. Could you point more precisely to the Implicit Function Theorem and its proof?
2. Eq. (11): Could something more be said about the properties of the approximation?

**Suggestions:**
- L105: "optimal solution" -> "optimal solution when $\epsilon=0$".
- Fig. 2: "Each contour" -> "Each set of concentric contours"?
- FastSOURCE could be explained more clearly, with references to equations.
- An overview of computation time for different TDA and evaluation methods would be interesting.

**Limitations:**

The limitations seem to be addressed adequately.

---

> ### Author Rebuttal · Authors · 2024-08-06
>
> We appreciate the reviewer's positive assessment of our paper, acknowledging that it is well-written, technically sound, and presents novel insights. We are grateful for your attention to detail in identifying errors and typos, which we will address in the revised manuscript.
>
> > **Could you point more precisely to the Implicit Function Theorem and its proof?**
>
> For a detailed derivation of influence functions using the Implicit Function Theorem, we direct the reviewer to [1] (Appendix B.1). With the assumptions outlined in our paper (the parameters are optimized to convergence and the objective has a unique solution), we have:
>
> \begin{align}
> \mathbf{0} = \nabla \mathcal{J}(\theta^\star) + \frac{\epsilon_0}{N} \nabla \mathcal{L}(\theta^\star, z_m),
> \end{align}
>
> where $\epsilon_0 = 0$. By the IFT, there exists a unique continuously differentiable function $r$ defined in the neighborhood of $\epsilon_0$ such that:
> \begin{align}
> \mathbf{0} = \nabla \mathcal{J}(r(\epsilon)) + \frac{\epsilon}{N} \nabla \mathcal{L}(r(\epsilon), z_m).
> \end{align}
>
> Taking the derivative with respect to $\epsilon$ at 0, we obtain:
> \begin{align}
> \mathbf{0} = \nabla^2 \mathcal{J}(\theta^\star) \frac{\mathrm{d} r}{\mathrm{d} \epsilon} \Big\vert_{\epsilon=0} + \frac{1}{N} \nabla \mathcal{L}(\theta^\star, z_m).
> \end{align}
>
> Note that $r(0) = \theta^\star$. Rearranging the terms yields the expression in Equation (4). We will explicitly mention this in the updated manuscript.
>
> > **Eq. (11): Could something more be said about the properties of the approximation?**
>
> In Equation (11), we approximate the Jacobians of different segments as statistically independent. There are two sources of randomness (as described in our footnote): (1) mini-batch sampling, which contributes to independence, and (2) autocorrelations in the optimization step, which induce correlations between optimization steps. Our approximation neglects the latter correlation.
>
> > **Re: Suggestions**
>
> We appreciate your suggestions and will incorporate them in the next revision of our manuscript.
>
> [1] Bae, J., Ng, N., Lo, A., Ghassemi, M., & Grosse, R. B. (2022). If influence functions are the answer, then what is the question?. Advances in Neural Information Processing Systems, 35, 17953-17967.

---

> > ### Comment · Reviewer_5ybi · 2024-08-11
> > **Thank you for the clarifications**
> >
> > Thank you for the clarifications!
> >
> > Taking everything into account, I think that I will keep my positive evaluation of the paper.

---

### Official Review · Reviewer_3Y11 · 2024-07-11

**Soundness:** 3
**Presentation:** 3
**Contribution:** 2
**Rating:** 5
**Confidence:** 4

**Summary:**

This paper propose a new TDA method (SOURCE) that combines implicit-differentiation-based methods and unrolling-based approaches. The new method is an extension to the SGD-influence and inherits the advantage to support multi-stage training pipelines while reduce the computation complexity to store and calculate the hessian matrix for each optimization step. The paper also carried out several experiments to show that SOURCE outperforms existing TDA methods under several training settings.

**Strengths:**

- The paper is well presented.

	○ The math derivation is self-contained, section 3.1 is a restate of SGD-influence and the extension to all iterations, 3.2 for segmentation and approximation, which is easy to follow.

	○ Some figures (especially figure 3) help me to understand the algorithm in a more intuitive way.

- The contextual information of this paper in the whole TDA community is stated clearly.

	○ It's stated clearly about the relationship between IF/SGD-Influence/SOURCE.

- The experiments for different training settings are convincing.

	○ This including 4 normal settings, 2 non-converging settings, 2 multi-stage training settings.

- The segmentation trick is quite intuitive and reasonable.

**Weaknesses:**

- First of all, I am not quite convinced and identify difference between the segmentation proposed in this paper and the one in TracIN (and also using different checkpoints in one training process for IF/TRAK, as stated in the paper as implicit-differentiation-based methods).

	○ Furthermore on this point, it will be good if the experiment for IF/TRAK could also include a version which ensembles (natively for IF and term-wisely for TRAK) on multiple checkpoints for each independently trained models.

	○ TRAK also made some experiment to show that it can use non-converge checkpoints to get a comparable LDS, could be mentioned in the paper as well.

- Not quite sure how much performance (accuracy) degradation is involved by the segmentation since there is no comparation directly with sgd-influence.

**Questions:**

1. **I am willing to make the rating higher** if the first weakness is resolved (intuitively or mathematically will be enough) since I understand add new experiment might be time-consuming.

**Limitations:**

I believe there is no negative societal impact of this work.

---

> ### Author Rebuttal · Authors · 2024-08-06
>
> We greatly thank the reviewer for their thorough evaluation and insightful comments. We are pleased that our paper's presentation and experimental results were well-received. We will address your concerns and questions below.
>
> > **I am not quite convinced and identify the difference between the segmentation proposed in this paper and the one in TracIn.**
>
> The key distinction lies in the quantities that TracIn and Source (or unrolled differentiation) approximate. As detailed in Section 4, TracIn estimates the importance of a training data point by aggregating the total change in the query's measurable quantity with gradient updates from this data point throughout training:
>
> \begin{align}
> \tau_{\text{TracIn}} (z_q, z_m, \mathcal{D}) = \sum_{i \in \mathcal{C}} \eta_i \nabla f(\theta_i, z_q)^\top \nabla \mathcal{L}(\theta_i, z_m),
> \end{align}
>
> where $\mathcal{C}$ denotes the set of steps the data point $z_m$ appeared during training, and $\eta_i$ and $\theta_i$ denote the corresponding learning rate and parameters, respectively. Even with a zero Hessian assumption (as done in Hydra, which we compare against in Table 1), the unrolling formulation gives:
>
> \begin{align}
> \tau_{\text{Unrolling}} (z_q, z_m, \mathcal{D}) = \sum_{i \in \mathcal{C}} \eta_i \nabla f(\theta^s, z_q)^\top \nabla \mathcal{L}(\theta_i, z_m),
> \end{align}
>
> where $\theta^s$ is the final model parameter, and the measurement gradient is always computed on this parameter. In contrast to TracIn, observe that unrolling aims to approximate the change in the final measurable quantity due to the perturbation of the data point's weight. Schioppa et al. [1] noted that using more intermediate checkpoints for TracIn can even hurt the performance in counterfactual estimation. Our work is motivated from the unrolling perspective, and hence, approximates a different quantity from TracIn. If TracInCP's analog of Hydra is considered (which has not been considered before, to the best of our knowledge), it can be viewed as applying our stationary approximation only to the gradient. However, using Hessian is crucial for achieving good LDS performance and better TDA performance in general, as highlighted in our work and recent studies [2, 3].
>
> > **Using different checkpoints in one training process for IF/TRAK, as stated in the paper as implicit-differentiation-based methods.**
>
> To the best of our knowledge, the ensembling over checkpoints during the training process was primarily an empirical observation [2] rather than being theoretically motivated. From the unrolled differentiation perspective, our derivations recommend averaging the Hessian and gradient within a segment. This approach contrasts with the direct ensemble of multiple checkpoints. More importantly, the previous ensembling scheme does not provide a natural treatment for models trained in multiple stages or those that have not converged. Addressing these limitations of influence functions is the core contribution of our work. We acknowledge some additional caveats. Trak generally benefits from ensembling (even using the same checkpoint) due to the projection's randomness. Moreover, using slightly non-converged parameters can also be helpful in cases where the gradients are too small at the final checkpoint. We appreciate the reviewer's observation and will mention this in our next manuscript revision.
>
> In response to this feedback, we conducted additional experiments for different checkpoints in one training process for IF and TRAK. We considered two scenarios: (1) FashionMNIST and (2) FashionMNIST-C (not converged model). Similar to the findings from Park et al. [2], we observe an increase in the LDS when using multiple checkpoints from a single training run. However, Source still obtains higher LDS in general, and the discrepancy is larger, especially in settings where the model has not fully converged. Note that the relatively smaller improvements in FashionMNIST, where the models are trained with a fixed dataset for a sufficient number of iterations, are expected because of the close connections with influence functions, as described in line 209.
>
> **FashionMNIST:**
>
> | # Model / TDA Method | IF | IF Ensemble | Trak | Trak Ensemble | Source |
> |---|---|---|---|---|---|
> | 1 | 0.29 | 0.37 | 0.08 | 0.14 | **0.46** |
> | 10 | 0.45 | 0.48 | 0.26 | 0.35 | **0.52** |
>
> **FashionMNIST-C:**
>
> | # Model / TDA Method | IF | IF Ensemble | Trak | Trak Ensemble | Source |
> |---|---|---|---|---|---|
> | 1 | 0.34 | 0.36 | 0.06 | 0.08 | **0.63** |
> | 10 | 0.40 | 0.41 | 0.12 | 0.17 | **0.64** |
>
> > **Not quite sure how much performance (accuracy) degradation is involved by the segmentation since there is no comparation directly with sgd-influence.**
>
> SGD-Influence poses significant practical challenges: (1) it necessitates storing all intermediate checkpoints throughout the optimization process, (2) the method requires a series of Hessian-vector products (HVPs) and many Monte Carlo samples from multiple training runs, as described in Section 3.1. Unlike other gradient-based approaches, these computations cannot be parallelized efficiently; this process must be repeated for each training data point. (3) It is worth noting that the largest model considered in SGD-Influence was smaller than the MLP (MNIST) models used in our study. These factors make SGD-Influence computationally prohibitive for models and datasets we considered, which is where our method, SOURCE, shows its strengths.
>
> [1] Schioppa, A., …, Zablotskaia, P. (2024). Theoretical and practical perspectives on what influence functions do.
>
> [2] Park, S. …, Madry, A. (2023). Trak: Attributing model behavior at scale.
>
> [3] Deng, J., …, Ma, J. (2024). Efficient Ensembles Improve Training Data Attribution.

---

> > ### Comment · Reviewer_3Y11 · 2024-08-13
> >
> > Thank you for the reply. I will keep my positive score.

---

### Official Review · Reviewer_hk4U · 2024-07-12

**Soundness:** 3
**Presentation:** 4
**Contribution:** 2
**Rating:** 3
**Confidence:** 4

**Summary:**

The article discusses the limitations of existing training data attribution (TDA) methods, which aim to estimate how a model's behavior would change if specific data points were removed from the training set. The authors propose a new method called Source, which combines the benefits of implicit-differentiation-based and unrolling-based approaches. Source is computationally efficient and suitable for cases where implicit-differentiation-based methods struggle, such as non-converged models and multi-stage training pipelines. Empirical results show that Source outperforms existing TDA techniques, particularly in counterfactual prediction, where implicit-differentiation-based approaches fall short.

**Strengths:**

1. The writing in this article is very clear and excellent, making it easy to read and follow.

2. The design of this article avoids the assumption made by Koh et al.'s estimator that the loss is a graph function.

**Weaknesses:**

See the question part.

**Questions:**

Regarding this article, as a practitioner in the field, I would like the authors to answer a series of questions.

1. We all hope that Data-attribution/Influence Analysis can find suitable application areas, such as the work done by Gross et al. in explaining the output of LLM. However, currently, we have not seen Data-attribution/Influence Analysis truly solve a series of real problems, especially for large models such as Stable-Diffusion/LLM/LVM.

2. If the time and computational cost required for attribution is much higher than that for training (as it should compute the gradient one-sample-by-one-sample, and even for many checkpoints), is there any practical application significance worth for such high complexities?

**Limitations:**

1. Error Bound: This article lacks an analysis of the error bound for the estimator.

2. Regarding the contribution: The claim "it allows the attribution of data points at different stages of training" is not a unique contribution of this article, as some previous works [1,2,3] have achieved this without assuming that a checkpoint with testing is necessarily at gradient=0. Therefore, I do not believe that this claim can be considered a significant contribution to this article.

3. Experiments: This article lacks validation on significant datasets (even without experiments on relatively old datasets like ImageNet) and instead tests on very small toy datasets.

[1]. S Hara, et al. Data Cleansing for Models Trained with SGD. NeurIPS.

[2]. G Pruthi, et al.  Estimating Training Data Influence by Tracing Gradient Descent. NeurIPS.

[3]. H Tan, et al. Data pruning via moving-one-sample-out. NeurIPS.

---

> ### Author Rebuttal · Authors · 2024-08-06
>
> We thank the reviewer for their thorough evaluation and insightful comments. Our research addresses limitations of influence functions and proposes a practical and novel algorithm to overcome these challenges. Grosse et al. [8], which the reviewer cited as an example of a TDA application, highlight a limitation of their approach:
>
> *"A second limitation is that we focus on pretrained models. Practical usefulness and safety of conversational AI assistants depend crucially on fine-tuning …. Extending influence functions … to the combination of pretraining and fine-tuning is an important avenue to explore."*
>
> Our work takes a step towards addressing this challenge. While we appreciate the reviewer's critique, we believe some concerns may not fully align with our research's core contributions. The following sections provide detailed responses to address these concerns and clarify our work’s significance. We welcome any further discussion during the reviewer-author period.
>
> > **Have not seen TDA truly solve real problems…**
>
> Training examples undeniably play a significant role in shaping model behavior. [1] and [2] demonstrated that removing less than 1% of the CIFAR-10 and ImageNet training dataset (identified by TDA) can lead to misclassifying a substantial fraction of test images. Several works show the importance of training data for LLMs [3, 4] and diffusion models [5].
>
> TDA techniques have already shown promise in several areas:
> 1. **Data Curation:** TDA methods have proven helpful in increasing compute efficiency [6] and improving subgroup robustness [7]. Notably, [6] demonstrated that data curation with TDA techniques provides a 2x compute multiplier on standard LM problems compared to baseline techniques.
> 2. **Understanding Model:** [8] employed influence functions to study LLM generalization patterns. Their analysis uncovered surprising sensitivities to word ordering. Berglund et al. [9] later empirically verified this phenomenon, which they termed the "Reversal Curse."
> 3. **Economic Models:** Building a royalty model to compensate data providers properly is an important economic and societal question. Using TDA techniques, [10] and [11] developed initial prototypes for such royalty models.
>
> TDA remains an active and valuable area of research, as evidenced by these applications and empirical observations on the importance of training data. While TDA may not yet be fully integrated into practical pipelines (at least publicly), *we believe this does not diminish the importance of researching potential failure modes in existing TDA methods and developing more accurate techniques, such as our work*. Our research contributes to the ongoing effort to make TDA methods more reliable and applicable to real-world scenarios.
>
> > **Is TDA worth it?**
>
> Naive gradient-based TDA methods do require gradient computations for all data points, which is as expensive as one-epoch training. We would like to clarify several points. First, in our experiments, attributing a batch of query data points is much faster than re-training a single model, as our models were trained over multiple epochs. Secondly, the one-by-one gradient computation described by the reviewer can be significantly optimized. Modern libraries such as Functorch or Opacus allow for efficient per-sample gradient computation. Lastly, some approaches, like [10], achieved computation times significantly faster than one-epoch training. Source is designed with flexibility in mind and can readily incorporate these efficiency improvements (Appendix E.3).
>
> On a separate note, we have recently verified that our implementation runs on the Llama-3-8B model (OpenWebText) using academic resources. We will release this implementation.
>
> > **Lacks error bound for the estimator.**
>
> Investigating the error bound under weak/practical assumptions presents an interesting direction for future research. Note that state-of-the-art techniques, such as Trak [12], EKFAC [8], and TracIn do not provide error bounds. We believe that the empirical results presented in our work provide convincing evidence of Source's advantages. Our evaluation uses state-of-the-art metrics [12] and covers multiple data modalities, demonstrating the efficacy of our approach across diverse scenarios.
>
> > **The claim “support multi-stage” is not a unique contribution ….**
>
> Our work does not claim to be the first to perform unrolled differentiation, as explained in the preceding paragraph. Rather, we present this capability as a key advantage of Source, particularly in comparison to influence functions. As other reviewers have noted, our contribution lies in the novel techniques we have developed to formulate and solve this problem. However, our work is the first to construct a set of experiments to verify the effectiveness of TDA techniques in the multi-stage training setup. Here, Source demonstrated large improvements: (1) over 5x improvement in LDS and (2) over 15x improvement in terms of subset removal evaluation, when compared to TracIn.
>
> > **Lacks large datasets**
>
> The primary challenge lies in evaluation. For context, our CIFAR-10 & ResNet-9 pipeline required training the network 2,000 times for LDS computation and up to 1,800 times per TDA technique in the subset removal evaluation. Developing more efficient yet reliable evaluation metrics remains an open problem. Despite these constraints, we conducted comprehensive experiments across a diverse range of tasks, datasets, and architectures (up to 110M parameters): (1) regression (Concrete, Parkinsons), (2) image classification (MNIST, FashionMNIST, CIFAR-10), (3) text classification (QNLI, SST2, RTE), language modeling (WikiText-2), and continual learning (RotatedMNIST and PACS). We also examined linear and logistic regression problems in Appendix G.4. Our quantitative experimental scope surpasses many previous and recent publications from academic labs [10, 11, 13] in terms of variety in data types, tasks, and model architectures.

---

> ### Author Response · Authors · 2024-08-06
> **References**
>
> [1] Ilyas, A., Park, S. M., Engstrom, L., Leclerc, G., & Madry, A. (2022). Datamodels: Predicting predictions from training data. arXiv preprint arXiv:2202.00622.
>
> [2] Singla, V., Sandoval-Segura, P., Goldblum, M., Geiping, J., & Goldstein, T. (2023). A Simple and Efficient Baseline for Data Attribution on Images. arXiv preprint arXiv:2311.03386.
>
> [3] Lee, K., Ippolito, D., Nystrom, A., Zhang, C., Eck, D., Callison-Burch, C., & Carlini, N. (2021). Deduplicating training data makes language models better. arXiv preprint arXiv:2107.06499.
>
> [4] Longpre, S., Yauney, G., Reif, E., Lee, K., Roberts, A., Zoph, B., ... & Ippolito, D. (2023). A pretrainer's guide to training data: Measuring the effects of data age, domain coverage, quality, & toxicity. arXiv preprint arXiv:2305.13169.
>
> [5] Carlini, N., Hayes, J., Nasr, M., Jagielski, M., Sehwag, V., Tramer, F., ... & Wallace, E. (2023). Extracting training data from diffusion models. In 32nd USENIX Security Symposium (USENIX Security 23) (pp. 5253-5270).
>
> [6] Engstrom, L., Feldmann, A., & Madry, A. (2024). Dsdm: Model-aware dataset selection with datamodels. arXiv preprint arXiv:2401.12926.
>
> [7] Jain, S., Hamidieh, K., Georgiev, K., Ilyas, A., Ghassemi, M., & Madry, A. (2024). Data Debiasing with Datamodels (D3M): Improving Subgroup Robustness via Data Selection. arXiv preprint arXiv:2406.16846.
>
> [8] Grosse, R., Bae, J., Anil, C., Elhage, N., Tamkin, A., Tajdini, A., ... & Bowman, S. R. (2023). Studying large language model generalization with influence functions. arXiv preprint arXiv:2308.03296.
>
> [9] Berglund, L., Tong, M., Kaufmann, M., Balesni, M., Stickland, A. C., Korbak, T., & Evans, O. (2023). The reversal curse: LLMs trained on" a is b" fail to learn" b is a". arXiv preprint arXiv:2309.12288.
>
> [10] Choe, S. K., Ahn, H., Bae, J., Zhao, K., Kang, M., Chung, Y., ... & Xing, E. (2024). What is Your Data Worth to GPT? LLM-Scale Data Valuation with Influence Functions. arXiv preprint arXiv:2405.13954.
>
> [11] Deng, J., & Ma, J. (2023). Computational copyright: Towards a royalty model for ai music generation platforms. arXiv preprint arXiv:2312.06646.
>
> [12] Park, S. M., Georgiev, K., Ilyas, A., Leclerc, G., & Madry, A. (2023). Trak: Attributing model behavior at scale. arXiv preprint arXiv:2303.14186.
>
> [13] Deng, J., Li, T. W., Zhang, S., & Ma, J. (2024). Efficient Ensembles Improve Training Data Attribution. arXiv preprint arXiv:2405.17293.

---

> > ### Comment · Reviewer_hk4U · 2024-08-14
> >
> > 1. Why don't you conduct experiments on LLMs/VLMs/SD models or even practical experiments on ImageNet? But just so many toy experiments?
> > 2. This work lacks theoretical justification.
> > 3. **First, in our experiments, attributing a batch of query data points is much faster than re-training a single mode.** So, the query data batch is a random batch?

---

> ### Author Response · Authors · 2024-08-14
>
> We appreciate the reviewer's follow-up questions. While we addressed questions 1 and 2 in our previous response, we are happy to provide further clarification. Please see our responses to your questions below.
>
> > **Why don't you conduct experiments on LLMs/VLMs/SD models or even practical experiments on ImageNet?**
>
> Replicating a setup similar to Grosse et al. (2023) [8] (e.g., LLMs) requires significant computational resources beyond most academic labs' reach. For instance, an experiment on a 13B parameter model, searching over 1M training sequences, would incur costs in the range of tens of thousands of dollars. Given sufficient resources, running more extensive experiments would be possible. As mentioned in our previous response, we recently confirmed that our implementation runs on the Llama3-8B model using academic resources.
>
> It is also worth noting that the primary issue lies in evaluation rather than performing data attribution on large models. Obtaining ground truth for counterfactual estimation is a computationally expensive process. It requires retraining the model up to 2,000 times for LDS and 10,000 times for the subset removal evaluation presented in Figure 5. While repeating these experiments on larger-scale datasets is possible, doing so would incur substantial costs that are difficult to conduct for most academic labs. Our paper demonstrates the effectiveness of our approach in various settings, including models with up to 110M parameters (e.g., BERT & QNLI dataset), which we believe are not just small toy experiments.
>
> > **This work lacks theoretical justification.**
>
> Section 3 of our paper provides a theoretical foundation, introduces a set of approximations needed to obtain practical algorithms, discusses their implications, and formally derives a tractable and efficient algorithm. To support our theoretical work, we offer empirical evidence demonstrating that our algorithm performs better than previous TDA techniques in classical (linear and logistic regression) and modern (neural networks) settings. Based on the previous review, we believe the reviewer argues that our work lacks theoretical justification solely because the error-bound analysis is not provided. It is important to note that state-of-the-art methods in the field, including EKFAC [8] and Trak [12], do not provide error-bound analyses yet are still theoretically justified. Establishing error-bound analysis under realistic assumptions, which are challenging in neural network settings, is an exciting direction for future research.
>
> > **Is the query data batch a random batch?**
>
> The query batch can consist of data points from the test dataset (e.g., 2,000 test data points of interest), which we use to compute influence scores across all training data points. For generative models, it could also consist of the model's output [10]. We are not entirely clear on what the reviewer meant by "random batch," as it could be just any batch of data points on which one would like to perform TDA. For context, our argument was that the reviewer's statement, “*the time and computational cost required for attribution is much higher than that for training*,” may not necessarily be true. Approximating leave-one-out (LOO) by retraining the model, as the reviewer described, requires multiple retraining for *each* training data point, which is significantly more computationally expensive.
>
> The discussion period ends in a couple of hours, but we would be happy to provide further clarification within the given timeframe if needed. Thank you again for your time and expertise.

---

### Official Review · Reviewer_9an4 · 2024-07-15

**Soundness:** 3
**Presentation:** 3
**Contribution:** 3
**Rating:** 7
**Confidence:** 4

**Summary:**

This paper introduces SOURCE, a new data attribution method designed primarily for deep neural networks; more generally, SOURCE is suited for any model class optimized with gradiend-based methods. The authors motivate SOURCE as an approximation to gradient unrolling, i.e., differentiating through the entire training process. They introduce a number of assumptions and simplifications that allow them to arrive at a formulation resembling existing influence-function-based data attribution methods.

The main assumption the authors employ is that the model Hessian (of the loss w.r.t. the model parameters) remains constant throughout segments of the training process (they set the number of segments $L$ as a hyperparameter). Additionally, the authors use the EK-FAC heuristic approximation for the Hessian [1] to reduce the computational cost of SOURCE. The same approximation has been employed in a previous data attribution method [2], which the authors use as a baseline.

In the experimental section of the paper, the authors evaluate SOURCE, alongside with an extensive list of baselines, on numerous (albeit rather small) datasets. SOURCE outperforms existing methods on the vast majority of benchmarks on standard data attribution evaluation metrics.

[1] T. George, C. Laurent, X. Bouthillier, N. Ballas, and P. Vincent. Fast approximate natural gradient descent in a kronecker factored eigenbasis. Advances in Neural Information Processing Systems, 31, 2018.

[2] R. Grosse, J. Bae, C. Anil, N. Elhage, A. Tamkin, A. Tajdini, B. Steiner, D. Li, E. Durmus, E. Perez, E. Hubinger, K. Lukošiu¯te˙, K. Nguyen, N. Joseph, S. McCandlish, J. Kaplan, and S. R. Bowman. Studying large language model generalization with influence functions, 2023.

**Strengths:**

on the selected datasets (Concrete, FashionMNIST, CIFAR-10, RTE, PACS), SOURCE performs consistently well, largely outperforming existing methods
- the paper is clearly written: the authors contextualize their method well within the existing literature and succinctly describe the novel ideas resulting in their method SOURCE
- the direction of adapting data attribution methods towards realistic (for deep learning) scenarios like lack of convergence and multi-stage training procedures is very timely and important; the paper does a great job at balancing between addressing these challenges and maintaining computational efficiency
- the resulting method, SOURCE, is easy to understand and is rather "obvious" (in hindsight, of course)

**Weaknesses:**

- compared to [1], SOURCE requires significantly larger amount of compute (as the number $L$ of training trajectory segment increases). At the same time, I would label the increase in performance from [1] to SOURCE as only "moderate". Given, in addition to the previous point, the conceptual similarity between the two methods, the technical contribution of this paper seems to be on the smaller side. Yet, I still appreciate the connection with unrolling that the authors present.
- if possible, it would be good to see (at least a qualitative) evaluation of SOURCE on a larger scale setting, especially since the mechanically very similar method [1] is employed on larger-scale language models; this said, I understand that the rebuttal period is short, and would not base my score change on this result.

[1] R. Grosse, J. Bae, C. Anil, N. Elhage, A. Tamkin, A. Tajdini, B. Steiner, D. Li, E. Durmus, E. Perez, E. Hubinger, K. Lukošiu¯te˙, K. Nguyen, N. Joseph, S. McCandlish, J. Kaplan, and S. R. Bowman. Studying large language model generalization with influence functions, 2023.

**Questions:**

- can the authors provide an estimate of the computational cost of each method (e.g. wall-clock time) used in Figure 4? My understanding is that SOURCE uses up to 6x the amount of compute as the reported baselines (line 283: "TRACIN and SOURCE use at most 6 intermediate checkpoints saved throughout training."). More generally, can the authors provide an equivalent to Figure 4 where compute, rather than the number of checkpoints, is equalized?
- how does the performance (e.g., as measured by LDS) of SOURCE scale as 1) the number of segments $L$ increases, and 2) the number of fully-retrained checkpoints increase? In particular, I am curious where LDS asymptotes, and whether the asymptote value for SOURCE is higher than the one for [1, 2]; if that is indeed the case, this would, in my opinion, significantly strengthen the argument that SOURCE has a smaller "modeling" error compared to previous approaches.
- my understanding is that the authors of [1] also propose using intermediate checkpoints along the training trajectory (without motivating this choice as an approximation to gradient unrolling); could the authors comment on the differences between the two estimators (excluding, obviously, the choice of Hessian approximation---EK-FAC and random projections, respectively)? Additionally, can the authors compare the performance of TRAK when intermediate checkpoints are used?

[1] S. M. Park, K. Georgiev, A. Ilyas, G. Leclerc, and A. Madry. TRAK: Attributing model behavior at scale. In International Conference on Machine Learning, pages 27074–27113. PMLR, 2023.

[2] R. Grosse, J. Bae, C. Anil, N. Elhage, A. Tamkin, A. Tajdini, B. Steiner, D. Li, E. Durmus, E. Perez, E. Hubinger, K. Lukošiu¯te˙, K. Nguyen, N. Joseph, S. McCandlish, J. Kaplan, and S. R. Bowman. Studying large language model generalization with influence functions, 2023.

---

> ### Author Rebuttal · Authors · 2024-08-06
>
> We thank the reviewer for their comprehensive and insightful evaluation of our work. We particularly appreciate the recognition of Source’s consistent performance across datasets, the clarity of our paper, and our efforts to balance addressing challenges in realistic scenarios while maintaining computational efficiency.
>
> > **The increase in performance as only moderate.**
>
> The moderate improvements in the first set of experiments, where models are trained to near convergence with a fixed dataset, are expected because of the close connections with influence functions, as described in line 209. The primary advantage of our approach becomes evident when the assumptions of implicit-differentiation-based methods fall short. We demonstrate this in two scenarios: models trained for only a few iterations and multi-stage training processes. In these scenarios, Source shows large improvements over influence functions, both in terms of LDS and subset removal evaluation.
>
> > **If possible, it would be good to see (at least a qualitative) evaluation of SOURCE on a larger scale setting …**
>
> We appreciate the reviewer's suggestion and agree that such an evaluation would provide useful insights. However, replicating a setup similar to [1] requires significant computational resources beyond most academic labs' reach. For instance, an experiment on a 13B parameter model, searching over 1M sequences, would incur costs in the range of tens of thousands of dollars. However, we have recently verified our EKFAC implementation scales to Llama-3 8B models. We will release our code to support TDA on larger-scale settings.
>
> > **Computational cost of each method (e.g., wall-clock time)**
>
> We have recently improved the efficiency of our EKFAC implementation and subsequently re-ran a subset of the experiments. While we have implemented additional optimization tricks such as query batching [2] and automatic mixed precision — which can significantly improve efficiency with only a slight decrease in LDS — we will report the wall-clock time without these improvements to maintain consistency with the presented results. It is important to note that (1) the wall-clock time includes model training time, and (2) although we used 2,000 query data points in our experiments, consistent with [1], Trak becomes significantly more computationally efficient as the number of query data points increases, as it does not require multiple iterations through the dataset. While we demonstrated one instantiation of Source with EKFAC, we emphasize that Source can also be implemented with Trak, as detailed in Appendix E.3.
>
> FashionMNIST
>
> | Method | TracIn (10 Models) | Trak (1 Model) | Trak (10 Models) | IF (1 Model) | IF (10 Models) | Source (1 Model) | Source (10 Models) |
> |---|---|---|---|---|---|---|---|
> | Time (s) | 361.22 | 25.40 | 138.66 | 26.73 | 267.91 | 50.55 | 495.39 |
> | LDS | 0.21 | 0.08 | 0.26 | 0.30 | 0.45 | 0.46 | 0.53 |
> | Frac. Misclassified Test Examples (300) | 0.23 | 0.10 | 0.36 | 0.30 | 0.44 | 0.54 | 0.65 |
>
> FashionMNIST-C
>
> | Method | TracIn (10 Models) | Trak (1 Model) | Trak (10 Models) | IF (1 Model) | IF (10 Models) | Source (1 Model) | Source (10 Models) |
> |---|---|---|---|---|---|---|---|
> | Time (s) | 115.26 | 7.58 | 32.72 | 8.88 | 89.68 | 18.29 | 182.93 |
> | LDS | 0.48 | 0.07 | 0.12 | 0.34 | 0.40 | 0.63 | 0.64 |
> | Frac. Misclassified Test Examples (300) | 0.56 | - | 0.18 | - | 0.41 | 0.90 | 0.91 |
>
> CIFAR-10
>
> | Method | TracIn (10 Models) | Trak (1 Model) | Trak (10 Models) | IF (1 Model) | IF (10 Models) | Source (1 Model) | Source (10 Models) |
> |---|---|---|---|---|---|---|---|
> | Time (s) | 3843.74 | 247.41 | 1843.11 | 232.15 | 2323.6 | 886.15 | 8896.95 |
> | LDS | 0.06 | 0.05 | 0.24 | 0.13 | 0.20 | 0.18 | 0.22 |
> | Frac. Misclassified Test Examples (1400) | 0.28 | 0.11 | 0.41 | 0.26 | 0.41 | 0.52 | 0.61 |
>
> > **LDS asymptotes**
>
> We direct reviewers to Figure 4 for insights on performance with different numbers of segments. In our experiments with the FashionMNIST pipeline, we did not observe an improvement in the LDS as we increased the number of segments to $L=6$. This may be because the segmentation approximation is already sufficiently accurate for these simpler experiments. We have conducted additional experiments to address this aspect of the reviewers' question about scaling performance with the number of fully re-trained checkpoints. We have trained up to 200 models and ensembled the models accordingly for FashionMNIST and FashionMNIST-C datasets. The uploaded PDF shows the results (we also compare it against the wall-clock time). On the FashionMNIST dataset, the gap between EKFAC and Source reduces as the number of checkpoints increases, but we still obtain a higher LDS at asymptote. On FashionMNIST-C (not converged), Source still obtains a significantly high LDS even at asymptote. We are conducting experiments of Trak with at most 300 models and we will include in the next revision of the manuscript.
>
> > **Re: Intermediate checkpoints**
>
> Our derivations recommend averaging the Hessian and gradient within a segment. This contrasts with the direct ensemble of multiple checkpoints explored in [1]. A core contribution of our work is the treatment for models trained in multiple stages or those that have not fully converged. Regarding the differences, Trak, as used in [1], benefits more from ensembling due to the randomness in the projection. We have observed that Trak can achieve higher performance even when ensembling a single checkpoint, while we generally do not see this improvement with EKFAC. To illustrate the performance differences, we have conducted additional experiments on Trak with intermediate checkpoints ensembling for FashionMNIST and FashionMNIST-C datasets (please see our response to Reviewer 3Y11; omitted due to characters limit). We will include additional datasets with intermediate checkpoints ensembled Trak in the next revision of the manuscript.

---

> > ### Comment · Reviewer_9an4 · 2024-08-13
> > **Score update**
> >
> > I appreciate the response and additional experiments. As a result, I am raising my score.

---

### Author Rebuttal · Authors · 2024-08-06

Dear Reviewers,

We thank all reviewers for their detailed and thoughtful feedback. We are pleased that the reviewers found our work to be well-written, easy to read (**9an4**, **hk4U**, **3Y11**, **5ybi**), addressing an important problem in the field (**9an4**, **3Y11**), technically solid (**9an4**, **5ybi**), and supported by convincing experiments (**3Y11**, **5ybi**).

We will address your concerns and questions in individual responses and are committed to improving our paper based on your valuable insights. Should you have additional comments, we welcome further discussion during the author-reviewer period. Once again, thank you for your time and expertise.

Regards,

Authors of "Training Data Attribution via Approximate Unrolling"

---

### Decision · Program_Chairs · 2024-09-25

**Decision:**

Accept (poster)

**Comment:**

The authors present a training data attribution method based on implicit differentiation and unrolling. There were four reviews, with three reviews fairly clear in their vote for accept. One reviewer voted for rejection, but did not make a compelling case when compared with the other three. All four reviewers agreed that the paper is in a well-polished state that makes the technical ideas clear as would be expected by a NeurIPS audience. Some concerns were voiced over the comparatively small size of the data sets, which could limit the potential usefulness of the optimization approach in the largest settings, but the results shown were convincing that the paper presents a useful idea.